# AOT*: Efficient Synthesis Planning via LLM-Empowered AND-OR Tree Search

## Abstract

Retrosynthesis planning enables the discovery of viable synthetic routes for target molecules, playing a crucial role in domains like drug discovery and materials design. Multi-step retrosynthetic planning remains computationally challenging due to exponential search spaces and inference costs. While Large Language Models (LLMs) demonstrate chemical reasoning capabilities, their application to synthesis planning faces constraints on efficiency and cost. To address these challenges, we introduce AOT*, a framework that transforms retrosynthetic planning by integrating LLM-generated chemical synthesis pathways with systematic AND-OR tree search. To this end, AOT* atomically maps the generated complete synthesis routes onto AND-OR tree components, with a mathematically sound design of reward assignment strategy and retrieval-based context engineering, thus enabling LLMs to efficiently navigate in the chemical space. Experimental evaluation on multiple synthesis benchmarks demonstrates that AOT* achieves SOTA performance with significantly improved search efficiency. AOT* exhibits competitive solve rates using 3-5× fewer iterations than existing LLM-based approaches, with the performance advantage becoming more pronounced on complex molecular targets. Our code is available at `https://anonymous.4open.science/r/AOTstar-31FD/`.

## 1 Introduction

Retrosynthetic planning, the decomposition of target molecules into commercially available building blocks, is a fundamental challenge in organic chemistry that requires navigating an exponentially growing search space of chemical transformations (Corey & Wipke, 1969; Nicolaou & Chen, 2009; Grzybowski et al., 2009; Lewell et al., 1998). While early rule-based expert systems demonstrated the feasibility of computer-aided synthesis planning (CASP), they suffered from extensive manual curation requirements and brittle performance on novel molecular scaffolds (Law et al., 2009; Boda et al., 2007; Coley et al., 2017). The advent of deep learning has enabled neural networks to automatically learn chemical transformations from large reaction databases, achieving remarkable progress in single-step reaction prediction (Segler et al., 2018; Schwaller et al., 2019; Segler & Waller, 2017; Liu et al., 2017; Schwaller et al., 2020; Dai et al., 2019; Chen & Jung, 2021). However, extending these successes to multi-step synthesis planning remains computationally challenging, as it requires sophisticated search strategies to efficiently explore the combinatorial space while maintaining reaction feasibility and synthetic accessibility (Christ et al., 2012; Bøgevig et al., 2015; Genheden et al., 2020; Saigiridharan et al., 2024; Thakkar et al., 2021; Tu et al., 2025; Dong et al., 2022).

Current neural approaches to multi-step synthesis planning face several challenges that limit their practical deployment (Maziarz et al., 2025; Genheden & Bjerrum, 2022). First, the computational overhead of repeated neural network inference creates significant bottlenecks, particularly problematic for high-throughput screening applications where thousands of molecules must be evaluated within tight time constraints (Andronov et al., 2025; Zhao et al., 2024; Hong et al., 2023). Second, these methods require extensive high-quality training data of validated synthesis routes to learn effective search strategies, yet when data is insufficient, they may exhibit limited performance and bias toward well-explored chemical spaces (Lin et al., 2022; Liu et al., 2023; Kim et al., 2021; Tripp et al., 2023; Yu et al., 2024). Third, the tree search algorithms underlying multi-step planning frequently suffer from redundant explorations and limited generalization beyond their training distributions, as

they cannot leverage broader chemical knowledge without explicit supervision (Kishimoto et al., 2019; Chen et al., 2020; Hong et al., 2023; Zhao et al., 2024).

The recent emergence of Large Language Models (LLMs) has opened new frontiers in chemical informatics, offering unprecedented capabilities for chemical reasoning (Boiko et al., 2023; White et al., 2023; Jablonka et al., 2024; M. Bran et al., 2024; Jablonka et al., 2024; Mirza et al., 2025). Recent work has demonstrated that LLMs can achieve remarkable performance in single-step retrosynthesis prediction when augmented with domain-specific fine-tuning or reasoning capabilities (Edwards et al., 2022; Liu et al., 2025; Yang et al., 2024; Zhang et al., 2024; 2025a; Lin et al., 2025; Deng et al., 2025). Pioneer efforts in LLM-based multi-step planning have emerged, such as the LLM-Syn-Planner framework (Wang et al., 2025), which employs evolutionary algorithms with mutation operators to generate and optimize complete retrosynthetic routes (Bran et al., 2025). However, extending these successes to practical multi-step synthesis planning remains challenging due to the computational expense of LLM inference, limited search efficiency with constrained iteration budgets, and the difficulty of incorporating chemical knowledge into the search process effectively (Guo et al., 2023; Kambhampati et al., 2024; Wang et al., 2024; Song et al., 2025).

To address these fundamental limitations, we introduce AOT*, a novel framework that harnesses the superior reasoning capabilities of LLMs while maintaining the computational efficiency required for practical synthesis planning (Jončev et al., 2025). Our approach builds upon the classical AND-OR tree representation of multi-step synthesis pathways, where OR nodes represent molecules and AND nodes represent reactions connecting products to their reactants (Chen et al., 2020; Schreck et al., 2019; Shi et al., 2020; Somnath et al., 2021). The key innovation of AOT* lies in its systematic integration of pathway-level LLM generation with AND-OR tree search, where complete synthesis routes are atomically mapped to tree structures, enabling efficient exploration through intermediate reuse and structural memory that reduces search complexity while preserving the strategic coherence of generated pathways.

Our contributions are threefold: (1) We present AOT*, a framework that integrates LLM-generated synthesis pathways with AND-OR tree search, enabling systematic exploration by atomically mapping pathways to tree structures that preserves synthetic coherence while exploiting structural reuse. (2) We demonstrate 3-5× efficiency improvements over existing approaches, with particularly strong performance on complex molecular targets where the tree-structured search effectively navigates challenging synthetic spaces that require sophisticated multi-step strategies. (3) We show consistent performance gains across diverse LLM architectures and benchmark datasets, confirming that the efficiency advantages stem from the algorithmic framework rather than model-specific capabilities, enabling practical deployment under various computational constraints.

## 2 RELATED WORK

### 2.1 SEARCH FOR RETROSYNTHESIS PLANNING

Multi-step retrosynthesis planning leverages search algorithms to discover complete synthetic pathways. Monte Carlo Tree Search (MCTS) (Segler et al., 2018; Segler & Waller, 2017) pioneered neural-guided synthesis planning, with variants including Experience-Guided MCTS (Hong et al., 2023), hybrid MEEA combining MCTS with A* search (Zhao et al., 2024), and alternatives like Nested Monte Carlo Search and Greedy Best-First Search (Roucairol & Cazenave, 2024). The Retro* algorithm (Chen et al., 2020) introduced AND-OR tree representations with neural-guided A* search (Schreck et al., 2019), leading to extensions including PDVN with dual value networks (Liu et al., 2023), self-improving procedures (Kim et al., 2021), uncertainty-aware planning (Tripp et al., 2023), depth-first proof-number search (Kishimoto et al., 2019), and double-ended search (Yu et al., 2024). Beyond tree search, recent approaches also employ beam search (Schwaller et al., 2020; Andronov et al., 2025), graph neural networks (Wang et al., 2023; Zhao et al., 2025), iterative string editing (Han et al., 2024), and neurosymbolic programming (Zhang et al., 2025c). Since retrosynthesis has broad applicability for molecular discovery, many platforms exist encompassing industrial (Bøgevig et al., 2015; Grzybowski et al., 2018) and open-source platforms (Genheden et al., 2020; Saigiridharan et al., 2024; Coley et al., 2017; Tu et al., 2025).

## 2.2 LLMs for Chemical Reasoning and Synthesis Planning

Large language models have demonstrated remarkable capabilities in encoding chemical knowledge and performing sophisticated reasoning about molecular properties and transformations (Edwards et al., 2022; White et al., 2023; Jablonka et al., 2024). These capabilities have been leveraged through various approaches including domain-specific fine-tuning (Yang et al., 2024; Zhang et al., 2024), instruction-tuning for chemical tasks (Lin et al., 2025), and development of experimental planning agents (Boiko et al., 2023; M. Bran et al., 2024; Wang et al., 2024). Transformer models like RSGPT (Deng et al., 2025) achieve strong performance through pre-training on billions of synthetic reactions. Recently, LLMs have been applied directly to multi-step synthesis planning. DeepRetro (Sathyanarayana et al., 2025) combines iterative LLM reasoning with chemical validation and human feedback, while RetroDFM-R (Zhang et al., 2025b) uses reinforcement learning to train LLMs for explainable retrosynthetic reasoning. Ma et al. (2025) construct knowledge graphs from literature for macromolecule retrosynthesis planning. The LLM-Syn-Planner framework (Wang et al., 2025) employs evolutionary algorithms to iteratively refine complete pathways.

# 3 Methodology

## 3.1 Problem Formulation

We formulate retrosynthetic planning as a generative AND-OR tree search problem as follows. Given a target molecule $t$ and a set of available building blocks $\mathcal{B}$, we seek to construct an AND-OR tree $\mathcal{T} = (\mathcal{V}, \mathcal{E})$ where OR nodes $v \in \mathcal{V}_{OR}$ represent molecules and AND nodes $a \in \mathcal{V}_{AND}$ represent reactions. Each OR node can have multiple child AND nodes (alternative reactions), while each AND node connects to its parent OR node (product) and child OR nodes (reactants). We employ a generative function $g : \mathcal{M} \times \mathcal{S} \to \mathcal{P}$ that maps molecules and retrieved similar synthesis routes to reaction pathways. Here, $\mathcal{M}$ denotes the space of molecules, $\mathcal{S}$ represents retrieved synthesis examples, and $\mathcal{P}$ is the space of multi-step pathways where each pathway $p = \langle r_1, ..., r_n \rangle$ consists of sequential reaction steps, with each $r_i = (P_i \to \{R_{i,1}, ..., R_{i,k_i}\})$ transforming a product molecule $P_i$ into a set of reactants $\{R_{i,1}, ..., R_{i,k_i}\}$ (denoted $R_i$ for brevity). The objective is to find a valid synthesis tree $\mathcal{T}^*$ satisfying:

$$\mathcal{T}^* \in \mathcal{T}_{valid} \quad \text{s.t.} \quad \forall v \in \text{Leaves}(\mathcal{T}^*), v \in \mathcal{B} \tag{1}$$

where $\mathcal{T}_{valid}$ denotes chemically valid trees and Leaves($\mathcal{T}$) refers to terminal OR nodes. To guide the search efficiently, we employ a cost function $C(\mathcal{T})$ encoding synthetic complexity. Generated pathways are mapped onto the tree as subgraphs $\mathcal{G}_p \subseteq \mathcal{T}$, maintaining consistency between the linear pathway structure and the hierarchical tree representation.

## 3.2 Pathway-to-Tree Mapping: Handling Structural Constraints

The mapping from LLM-generated linear pathways to AND-OR tree structures presents unique challenges that require careful algorithmic design. We formalize this as a tree construction problem with consistency constraints (Fontana, 1990). For a generated pathway $p$ with reaction steps $r_i$, we construct a subtree $\mathcal{G}_p \subseteq \mathcal{T}$ that maintains three principal constraints: (1) Each molecule maps to exactly one OR node in the tree, enforced through SMILES canonicalization (Weininger, 1988; O'Boyle, 2012): $\forall m_1, m_2 \in \mathcal{M} : \text{canon}(m_1) = \text{canon}(m_2) \Rightarrow \text{OR}(m_1) = \text{OR}(m_2)$. (2) Reaction mappings preserve parent-child relationships across pathway steps—when step $r_i$ decomposes molecule $m$ appearing in step $r_j$ ($j < i$), we enforce: $m \in R_j \land P_i = m \Rightarrow \text{AND}(r_i) \in \text{Children}(\text{OR}(m))$. (3) All generated reactions map to the tree, but orphaned steps targeting already-solved molecules are pruned: $\text{Map}(r_i) = \text{AND}(r_i)$ if $\neg\text{IsSolved}(P_i)$, otherwise $\emptyset$. The mapping algorithm processes pathways recursively, starting from the first step connected to the target and matching subsequent steps to unsolved molecules through canonicalized SMILES comparison. Invalid pathways are discarded during template-based validation while valid ones proceed to tree integration. Atomically mapping complete pathways to tree structures preserves the strategic coherence of LLM-generated routes, contrasting with incremental methods that expand individual reactions without global synthetic strategy.

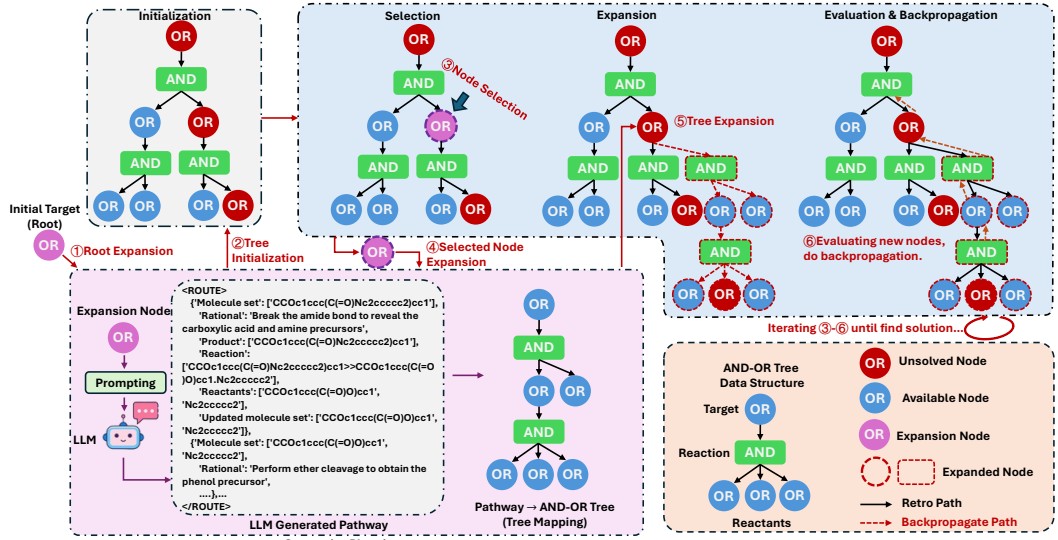

Figure 1: AOT* framework overview. The framework operates in four phases: (1) Initialization with root expansion via LLM-generated pathways, (2) Selection phase identifying promising nodes through exploration-exploitation balancing, (3) Expansion where selected OR nodes prompt LLM to generate multi-step pathways that are validated and mapped to tree structure, and (4) Evaluation and backpropagation to update node statistics. Blue circles indicate purchasable molecules, red circles represent unsolved targets, and green squares show AND reaction nodes. The generative process transforms LLM output into structured AND-OR tree branches while maintaining chemical validity.

## 3.3 AOT*: AND-OR TREE SEARCH WITH GENERATIVE EXPANSION

### 3.3.1 PATHWAY-LEVEL GENERATION FRAMEWORK

AOT* integrates LLM-based pathway generation with systematic AND-OR tree search. The framework transforms retrosynthetic planning through strategic generation of complete synthesis pathways guided by tree exploration. During node expansion, the framework prompts an LLM to generate complete multi-step synthesis routes for selected molecules: $\mathcal{P}_{\text{gen}} = \arg\max_{p \in \mathcal{P}} P(p \mid m, \mathcal{S}, \theta)$ where $m$ denotes the selected unsolved molecule, $\mathcal{S}$ represents retrieved similar synthesis routes, and $\theta$ parameterizes the LLM. The tree state $\mathcal{T}$ guides molecule selection but does not directly condition pathway generation. The generation process leverages the LLM's implicit chemical knowledge to propose routes that systematically reduce molecular complexity while maintaining synthetic feasibility. Each generated pathway decomposes the target through a sequence of transformations, producing complete routes. To incorporate chemical precedent, the framework employs retrieval-augmented generation (Lewis et al., 2020). For each selected molecule, structurally similar compounds are retrieved from a database of validated synthesis routes: $\mathcal{S}_{\text{similar}} = \text{top-}k_{s \in \mathcal{D}}\{\text{Tanimoto}(m, s)\}$ where similarity is computed using Tanimoto coefficient on Morgan fingerprints (Bajusz et al., 2015). These examples provide in-context demonstrations, guiding generation toward feasible strategies. The retrieved routes supply reaction precedents and strategic patterns while maintaining exploration flexibility. Generated pathways undergo template-based validation to verify chemical validity (Coley et al., 2019; Wang et al., 2025). Valid routes are mapped onto the AND-OR tree structure, creating subtrees that preserve pathway coherence. This mapping maintains both local reaction validity and global synthetic strategy consistency.

### 3.3.2 TREE SEARCH WITH GENERATIVE EXPANSION

Building upon the pathway-level generation framework, AOT* implements a systematic tree search framework that coordinates the exploration of the AND-OR tree structure with LLM generative expansion. The framework maintains exploration guarantees while leveraging the efficiency gains from pathway-level generation. The search process operates through four integrated phases:

**Selection Phase.** The selection procedure identifies the most promising leaf AND node for expansion utilizing the Upper Confidence Bound (UCB) criterion (Auer et al., 2002): $\text{UCB}(a) = \bar{v}_a + c\sqrt{\frac{\ln N_{\text{parent}}}{n_a}}$ where $\bar{v}_a$ denotes the empirical mean value computed from previous expansions, $N_{\text{parent}}$ represents the cumulative visitation count across sibling AND nodes, and $c$ constitutes the exploration-exploitation trade-off parameter. The selection mechanism targets expandable leaf AND nodes—those containing unsolved reactants and residing at depths below the predefined threshold—thereby allocating computational resources to the active search frontier.

**Expansion Phase.** Given a selected AND node $a$, the algorithm identifies constituent unsolved reactant molecules and generates synthesis pathways. When multiple unsolved reactants exist, the least-explored molecule is selected. The generative process employs an LLM conditioned on the selected molecule and retrieval-augmented examples: $p \sim P(p \mid v, \mathcal{S}(v), \theta)$ where $v$ denotes the selected molecule, $\mathcal{S}(v)$ represents retrieved similar synthesis routes, and $\theta$ parameterizes the LLM. For fair comparison considerations, we adopt the prompt design and RAG methodology from Wang et al. (2025). Detailed prompt templates and RAG implementation can be found in Appendix B.3. Generated pathways undergo template-based validation to ensure chemical feasibility. Valid pathways are mapped to the tree structure through a hierarchical construction process. For a generated pathway $p = \langle r_1, \ldots, r_n \rangle$ where $r_i = (P_i \rightarrow \{R_{i,1}, \ldots, R_{i,k_i}\})$, the algorithm constructs AND nodes for each reaction and OR nodes for each molecule:

$$\Psi(p) = \bigcup_{i=1}^{n} \left\{ \text{OR}(P_i) \xrightarrow{\text{AND}(r_i)} \{\text{OR}(R_{i,j})\}_{j=1}^{k_i} \right\} \tag{2}$$

This mapping generates subtrees where each AND node maintains parent-child relationships with corresponding OR nodes, preserving pathway coherence by connecting initial reactions to targets and recursively expanding unsolved intermediates.

**Evaluation Phase.** Generated AND nodes undergo evaluation via a composite reward function: $R(a) = \alpha \cdot f_{\text{avail}}(a) + (1-\alpha) \cdot f_{\text{chem}}(a)$ where $f_{\text{avail}}(a) \in [0, 1]$ quantifies the fraction of commercially available reactants, $f_{\text{chem}}(a) \in [0, 1]$ assesses chemical feasibility through synthetic complexity (SC) score evaluation (Coley et al., 2018), $\alpha$ is the availability-feasibility weight. This formulation balances synthetic accessibility with chemical viability.

**Backpropagation Phase.** Value estimates propagate through the tree structure following parent-child relationships: $\bar{v}_a^{(t+1)} = \frac{n_a \cdot \bar{v}_a^{(t)} + R(a_{\text{child}})}{n_a + 1}$. Upon molecular resolution (through commercial availability or complete synthesis), the solved status propagates throughout the tree. The algorithm marks solved OR nodes with corresponding solving AND paths, re-evaluates affected parent reactions, and prunes solved subtrees from the active search space. This update mechanism incorporates newly available intermediates across all branches of the search tree.

**Termination.** The search process terminates when: (i) a complete solution is found where $\forall v \in \text{Leaves}(\mathcal{T}), v \in \mathcal{B}$, (ii) computational budget limits are reached (maximum iterations), or (iii) the search space is exhausted with no remaining expandable nodes. Upon termination, the process returns either the complete synthesis tree or a partial solution with the most promising incomplete branches.

### 3.4 THEORETICAL ANALYSIS

Retrosynthetic planning requires searching through a combinatorial space with branching factor $b$ and depth $d$, resulting in $\mathcal{O}(b^d)$ complexity for exhaustive search. LLM-based methods using evolutionary algorithms operate through local mutations requiring $\mathcal{O}(\mu \cdot g)$ evaluations where $\mu$ is population size and $g$ is generations (Beyer & Schwefel, 2002; Wang et al., 2025). AOT* reduces this complexity to $\mathcal{O}(k \cdot d)$ where $k \ll b$ by replacing node-wise enumeration with pathway-level generation. The method leverages systematic tree search to explore the reduced search space. However, this approach inherits limitations from the exploration strategy employed. Let $q$ denote the LLM's generation quality—the fraction of generated pathways that are chemically valid and useful. When $q < 1$, we need approximately $1/q$ times more generations to find good solutions, giving effective complexity $\mathcal{O}(k/q \cdot d)$. Moreover, UCB only guarantees finding near-optimal solutions: the regret bound grows as $\mathcal{O}(\sqrt{n \log n})$ where $n$ is the number of expansions (Bubeck et al., 2012), meaning we cannot guarantee finding the truly optimal synthesis route. This transforms combinatorial opti-

mization into structured sampling from $P(p \mid m, \mathcal{T}, \theta)$ (Sun et al., 2023). Each LLM call explores a chemically-constrained subspace, achieving empirical efficiency gains of 3-5$\times$ over evolutionary methods (see Sec. 4.4 for details). The pathway-level coherence enables rapid convergence to good solutions, though performance fundamentally depends on LLM generation quality $q$.

## 4 EXPERIMENTS

### 4.1 EXPERIMENTAL SETUP

**Datasets.** We evaluate our methods on four retrosynthesis benchmarks. **USPTO-Easy** and **USPTO-190** (Chen et al., 2020) are derived from the USPTO dataset, containing 200 and 190 molecules respectively, with former representing simpler synthesis problems. **Pistachio Reachable (Pistachio Reach.)** and **Pistachio Hard** are from the Pistachio dataset [1], containing 150 and 100 molecules respectively, with Pistachio Hard presenting more challenging synthesis tasks. Following Wang et al. (2025), we use a route database constructed from training and validation sets of Retro* (no overlap with test molecules), while the reaction database is a processed version of USPTO-Full. We use 231 million purchasable compounds in eMolecules as building blocks (Chen et al., 2020).

**Baselines.** We compare against three categories of methods: (1) *Template-based search algorithms* including Graph2Edits (Zhong et al., 2023), RootAligned (Zhong et al., 2022), and Local-Retro (Chen & Jung, 2021) with both MCTS (Segler et al., 2018) and Retro* (Chen et al., 2020) search; (2) *Constrained Search (Constr.)* including DESP (Yu et al., 2024) using bidirectional search and Tango* (Jonček et al., 2025) guiding search towards specified starting materials; (3) *LLM-based approaches* (3) *LLM-based approaches* including (i) LLM (MCTS/Retro*) following Wang et al. (2025) where LLMs act as single-step reaction predictors using template selection and self-consistency sampling within traditional search algorithms; (ii) LLM-Syn-Planner (LLM-S.P.) (Wang et al., 2025) which employs evolutionary search to optimize the synthesis routes iteratively. For fair comparison, all methods use the same building block inventory and reaction templates. Notably, LLM-Syn-Planner (Wang et al., 2025) was provided with identical RAG and prompting strategies as AOT*, ensuring comparisons reflect algorithmic design rather than prompt engineering.

**Metrics.** We report *solve rate* (SR) as the primary metric, measuring the percentage of target molecules successfully synthesized within the search budget. We evaluate efficiency through: (1) *Solve rates* at multiple budgets: N (iterations)=100, 300, 500, to assess search efficiency; (2) *Iteration-to-solution* (Iters) analysis at fine-grained intervals: N=20, 40, 60, 80, 100, to measure convergence speed; (3) *Difficulty-stratified performance* by SC score (Coley et al., 2018) quartiles (Q1-Q4, from simplest to most complex) to examine efficiency across molecular complexity levels.

**Implementation Details.** To ensure fair comparison, we follow Wang et al. (2025) and evaluate AOT* using GPT-4o (Hurst et al., 2024) and DeepSeek-V3 (Liu et al., 2024) as the primary LLM models (We denote GPT-4o as "GPT" and DeepSeek-V3 as "DS" hereafter for brevity). We maintain main LLM configurations and prompts with Wang et al. (2025) to isolate algorithmic improvements. Framework-specific parameters include UCB exploration parameter $c = 0.5$, maximum search depth of 16 steps, and the availability-feasibility weight $\alpha = 0.4$; Throughout our experiments, N denotes the number of search iterations while n represents the number of RAG samples. Results reported in this section use 100 iterations (N = 100) as the default search budget unless otherwise specified.

### 4.2 MAIN RESULTS

Table 1 demonstrates AOT*'s superior efficiency in retrosynthetic search. At low computational budgets (N=100), AOT* achieves solve rates matching or exceeding competing methods' 500-iteration performance. On USPTO-190, AOT* (DS) reaches 93.1% at N=300, while LLM-Syn-Planner (DS) requires 500 iterations to achieve comparable performance (92.6%). This advantage is most pronounced on Pistachio Hard, where AOT* achieves 85-86% solve rates at N=100, while LLM-Syn-Planner requires 300-500 iterations to reach comparable performance (84-86%), demonstrating a 3-5$\times$ efficiency gain. Direct LLM integration (MCTS/Retro*) fails catastrophically with $\leq$ 5% solve rates, validating that pathway-level generation fundamentally outperforms single-step prediction. The performance gaps at N=100 (20% + on USPTO-190, 10% + on Pistachio Hard)

---

[1] https://www.nextmovesoftware.com/pistachio.html

Table 1: Comparison of solve rates (%) across different search budgets on four benchmark datasets. Best results are **bolded** and top-3 are underlined.

| | Method | USPTO-190 | | | Pistachio Hard | | | USPTO-Easy | | | Pistachio Reachable | | |
|---|---|---|---|---|---|---|---|---|---|---|---|---|---|
| | | N=100 | 300 | 500 | N=100 | 300 | 500 | N=100 | 300 | 500 | N=100 | 300 | 500 |
| Single-step | Graph2Edits (MCTS) | 42.7 | 54.7 | 63.5 | 26.0 | 41.0 | 62.0 | 90.0 | 93.5 | 96.5 | 77.3 | 88.4 | 94.2 |
| | RootAligned (MCTS) | 79.4 | 81.1 | 81.1 | 83.0 | 85.0 | 85.0 | 98.0 | 98.5 | 98.5 | 99.3 | **99.3** | **99.3** |
| | LocalRetro (MCTS) | 44.3 | 50.9 | 58.3 | 52.0 | 55.0 | 62.0 | 92.5 | 94.5 | 95.5 | 86.7 | 90.0 | 95.3 |
| | Graph2Edits (Retro*) | 51.1 | 59.4 | 80.0 | 71.0 | 74.0 | 82.0 | 92.0 | 95.5 | 97.5 | 94.0 | 95.0 | 97.5 |
| | RootAligned (Retro*)† | **86.8** | 88.9 | 88.9 | 78.0 | 82.0 | 82.0 | 99.0 | 99.0 | 99.0 | 98.7 | 98.7 | 98.7 |
| | LocalRetro (Retro*) | 51.0 | 65.8 | 73.7 | 63.0 | 69.0 | 72.0 | 95.5 | 97.5 | 98.0 | 97.3 | 99.3 | 99.3 |
| Constr. | DESP | 30.0 | 35.3 | 39.5 | 44.0 | 50.0 | – | – | – | – | 90.0 | 96.0 | – |
| | Tango* | 33.2 | 45.3 | 53.7 | 59.0 | 63.0 | – | – | – | – | 95.3 | 99.3 | – |
| LLM-based | LLM (MCTS) | 25.8 | 27.2 | 31.3 | 0.0 | 4.0 | 5.0 | 54.5 | 68.5 | 75.5 | 12.7 | 17.3 | 20.7 |
| | LLM (Retro*) | 23.2 | 26.8 | 30.6 | 0.0 | 2.0 | 5.0 | 56.0 | 69.0 | 75.5 | 14.7 | 19.3 | 13.3 |
| | LLM-Syn-Planner (GPT) | 64.7 | 91.1 | 92.1 | 72.0 | 86.0 | 87.0 | 91.0 | 99.5 | **100.0** | 93.3 | 98.0 | 98.0 |
| | LLM-Syn-Planner (DS) | 62.1 | 92.1 | 92.6 | 74.0 | 84.0 | 86.0 | 93.0 | 99.5 | **100.0** | 96.7 | 99.3 | 99.3 |
| Ours | AOT* (GPT) | 82.1 | 92.6 | 93.1 | 85.0 | 88.0 | **93.0** | 98.5 | **100.0** | **100.0** | 96.7 | 99.3 | 99.3 |
| | AOT* (DS) | 86.3 | **93.1** | **93.6** | **86.0** | **89.0** | **93.0** | **100.0** | **100.0** | **100.0** | 98.7 | 99.3 | 99.3 |

demonstrate our AND-OR tree's systematic exploration advantages over iterative evolutionary optimization. While template-based methods like RootAligned show limited gains (2.1% improvement from N=100 to N=500 on Pistachio Hard), AOT* with DeepSeek-V3 achieves +7.3% improvement, highlighting the generative approach's broader solution space.

## 4.3 DIFFICULTY-STRATIFIED PERFORMANCE ANALYSIS

Table 2 reveals that AOT*'s efficiency advantage increases with molecular complexity across all datasets. Both methods handle simple molecules (Q1) well, but AOT* generally requires 3-5× fewer iterations while maintaining comparable or better solve rates. On challenging datasets (USPTO-190 and Pistachio Hard), the performance gap becomes substantial at higher complexity. For Q4 molecules, LLM-Syn-Planner's solve rates drop to 27.6% and 56.0% respectively, while AOT* maintains 78.7% and 76.0%. De-

Table 2: Performance breakdown by SC score quartiles: AOT* *v.s.* LLM-Syn-Planner.

| | Method | USPTO Easy | | Pistachio Reach. | | Pistachio Hard | | USPTO-190 | |
|---|---|---|---|---|---|---|---|---|---|
| | | Iters | SR | Iters | SR | Iters | SR | Iters | SR |
| Q1 | LLM-S.P. | 10.33 | **100%** | 14.04 | **100%** | 34.83 | 92.0% | 33.06 | 91.6% |
| | AOT* | **2.78** | **100%** | **4.82** | **100%** | **5.76** | **100.0%** | **18.85** | **100.0%** |
| Q2 | LLM-S.P. | 28.10 | 98% | 23.81 | 97.3% | 36.50 | 80.0% | 35.86 | 74.4% |
| | AOT* | **9.10** | **100%** | **9.54** | **100%** | **13.92** | **88.0%** | **26.45** | **85.1%** |
| Q3 | LLM-S.P. | 31.68 | 92% | 26.58 | 93.3% | 47.11 | 68.0% | 41.18 | 54.1% |
| | AOT* | **10.26** | **100%** | **9.73** | **97.3%** | **28.68** | **80.0%** | **35.48** | **81.2%** |
| Q4 | LLM-S.P. | 44.67 | 82% | 27.75 | 93.3% | 56.60 | 56.0% | 45.79 | 27.6% |
| | AOT* | **15.65** | **100%** | **12.08** | **97.4%** | **32.92** | **76.0%** | **38.51** | **78.7%** |

spite using fewer iterations than LLM-Syn-Planner (38.51 vs 45.79 on USPTO-190), AOT* achieves nearly 3× better solve rates on the most complex targets. On simpler datasets (USPTO-Easy and Pistachio Reachable), both methods maintain high solve rates even for Q4 molecules, but AOT* still demonstrates superior efficiency. These demonstrate that AOT*'s tree-structured search scales better than evolutionary approaches, which suffer from redundant pathway exploration.

## 4.4 EFFICIENCY ANALYSIS

**Iteration Efficiency.** Table 3 demonstrates AOT*'s superior search efficiency across all benchmarks. With DeepSeek-V3, AOT* achieves 56.3% solve rate at 20 iterations on USPTO-190, surpassing LLM-Syn-Planner's performance at 60 iterations (46.8%). This efficiency gap is most pronounced on Pistachio Hard, where AOT* reaches 67.0% at 20 iterations while LLM-Syn-Planner achieves only 13.0%, representing a 5× improvement. Across all datasets, AOT* requires 3-5× fewer iterations to reach comparable solve rates, from 1.6× on simpler targets (USPTO-Easy) to over 5× on complex ones. This iteration efficiency stems from the AND-OR tree's ability to systematically exploit discovered intermediates and prune redundant branches, whereas LLM-Syn-Planner's evolutionary approach explores pathways independently without structural memory. The performance gains persist across both GPT-4o and DeepSeek-V3, with DeepSeek-V3 consistently slightly outperforming GPT-4o, confirming that our algorithmic framework effectively leverages diverse LLM capabilities.

Table 3: Solve rates (%) at different iteration thresholds.

| | | GPT | | DeepSeek | |
|---|---|---|---|---|---|
| Dataset | Iter. | LLM-S.P. | AOT* | LLM-S.P. | AOT* |
| Pistachio Hard | 20 | 9.0 | 64.0 | 13.0 | **67.0** |
| | 40 | 25.0 | 76.0 | 33.0 | **78.0** |
| | 60 | 50.0 | 79.0 | 55.0 | **81.0** |
| | 80 | 65.0 | 81.0 | 69.0 | **83.0** |
| | 100 | 72.0 | 85.0 | 74.0 | **86.0** |
| USPTO-190 | 20 | 9.5 | 55.7 | 10.5 | **56.3** |
| | 40 | 33.7 | 69.5 | 31.0 | **72.1** |
| | 60 | 52.6 | 78.4 | 46.8 | **81.6** |
| | 80 | 57.3 | 80.5 | 55.7 | **85.3** |
| | 100 | 64.7 | 82.1 | 62.1 | **86.3** |
| Pistachio Reach. | 20 | 65.0 | 84.7 | 66.7 | **87.3** |
| | 40 | 80.7 | 90.0 | 81.3 | **95.3** |
| | 60 | 85.3 | 94.0 | 88.0 | **97.3** |
| | 80 | 91.0 | 95.3 | 94.0 | **98.7** |
| | 100 | 93.3 | 96.7 | 96.7 | **98.7** |
| USPTO-Easy | 20 | 54.0 | 89.0 | 55.3 | **90.0** |
| | 40 | 71.3 | 93.5 | 72.0 | **94.5** |
| | 60 | 81.7 | 95.5 | 85.3 | **96.5** |
| | 80 | 88.3 | 96.5 | 90.0 | **99.0** |
| | 100 | 91.0 | 98.5 | 93.0 | **100.0** |

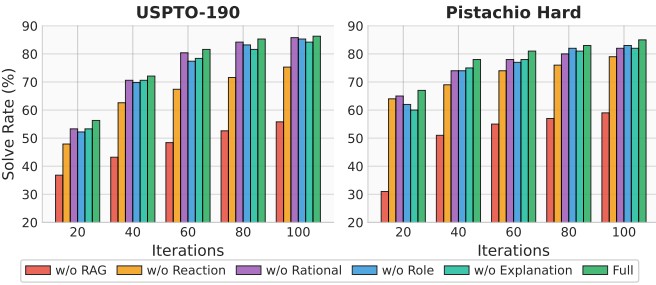

(a) Impact of prompt engineering strategies.

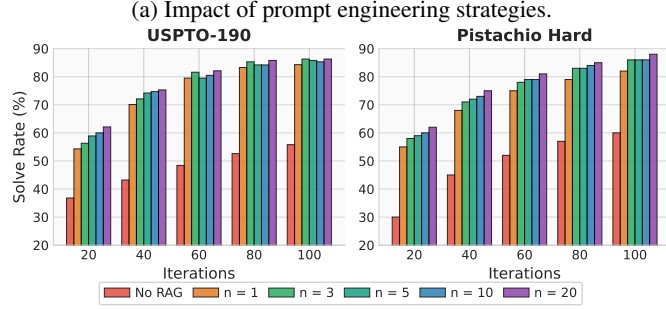

(b) RAG sample number (n) effects on search performance.

Figure 2: Component analysis on Pistachio Hard, USPTO-190.

**Component Ablation Analysis.** We further decompose the prompt into several components: role description, task description, planning requirement, explanation requirement, rational field, and detailed requirements parts. We conduct ablation studies on these prompt components together with RAG to analyze their individual contributions. Figure 2a and 2b reveal how each component contributes to AOT*'s search efficiency. RAG emerges as most critical, with its removal degrading solve rates by approximately 20 40% at early iterations and 20 30% at N=100. The method requires 2-3× more iterations for comparable performance without RAG. Optimal RAG configuration varies by target complexity: USPTO-190 saturates at 5 samples while Pistachio Hard continues improving to 10 samples, reflecting greater precedent requirements for complex natural products. Prompt engineering components (role, rationale, explanation) show modest individual impact but collectively accelerate search by 10-20 iterations. Their effect is most pronounced early (N=20-60) where AOT* establishes its efficiency advantage. These components work synergistically, with RAG providing chemical precedents, prompt engineering guiding exploration, and tree structure enabling intermediate reuse. This combination enables AOT* to identify viable synthesis routes 5-6× faster than evolutionary approaches lacking structural memory.

## 4.5 ABLATION STUDIES

**Hyperparameter Sensitivity.** Table 4 examines search hyperparameters on Pistachio Hard, revealing robust performance across configurations. The exploration parameter $c$ performs optimally at 0.5, achieving 84-86% solve rates across temperatures. Higher $c$ values yield diminishing returns, particularly when combined with high temperature (T= 0.9), where performance drops to 77% at $c = 1.414$. Temperature shows a sweet spot at 0.7 for $c$=0.5/1.0. The narrow performance range (77-86%) demonstrates AOT*'s stability—even suboptimal settings maintain reasonable solve rates. The best configura-

Table 4: Hyperparameter sensitivity analysis on Pistachio Hard.

| $c$ | Temperature | | | |
|---|---|---|---|---|
| | 0.3 | 0.5 | 0.7 | 0.9 |
| 0.5 | 84.0 | 85.0 | **86.0** | 85.0 |
| 1.0 | 83.0 | 83.0 | 84.0 | 79.0 |
| 1.414 | 84.0 | 80.0 | 80.0 | 77.0 |
| 2.0 | 83.0 | 83.0 | 82.0 | 78.0 |

tion ($c = 0.5$, T= 0.7) achieves 86%, only marginally better than alternatives, indicating relatively modest tuning requirements for practical deployment. Lower exploration parameters consistently outperform higher ones, suggesting LLM-generated pathways provide sufficient diversity without aggressive exploration.

**RAG Samples *v.s.* Token Usage.** Figure 3 quantifies the trade-off between retrieval-augmented generation effectiveness and computational cost. The analysis reveals a sharp performance plateau after 3 samples: solve rate increases from 60% (No RAG) to 86% (3 samples), then remains nearly flat despite token usage continuing to grow exponentially. Specifically, increasing from 3 to 20 samples yields only 2% performance gain while inflating token consumption by 177% (from 1,478 to 4,091 tokens). This diminishing returns pattern validates our default configuration of 3 samples, which achieves 86% solve rate on Pistachio Hard while using only 36% of the tokens required at 20 samples. The rapid saturation suggests that a small set of high-quality chemical precedents suf-

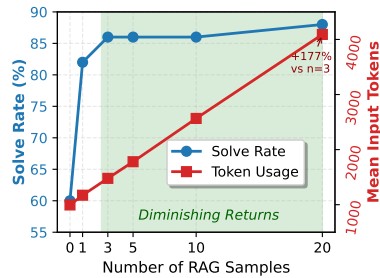

Figure 3: Performance saturation and input token usage with varying RAG samples on Pistachio Hard.

ficiently grounds the LLM's pathway generation, with additional examples providing redundant information rather than novel strategic insights.

### 4.6 CROSS-MODEL CONSISTENCY AND COST-PERFORMANCE TRADE-OFFS

Table 5 demonstrates AOT*'s efficiency and cost-performance advantage across diverse LLM architectures. Cost-performance analysis reveals that budget models achieve cost-competitive results: GPT-4o-mini ($0.15/$0.60 per million tokens) reaches 32% solve rate at N=20, while premium models like Claude-4-Sonnet ($3.00/$15.00) achieve 63% despite 20× higher costs. DeepSeek-V3 emerges as the optimal choice, achieving 67% at N=20 and 86% at N=100 with moderate pricing ($0.56/$1.68), matching or exceeding expensive alternatives.

Table 5: Performance across LLM architectures on Pistachio Hard. Cost: $/1M tokens.

| Model | Solve Rate (%) | | API Cost | |
|---|---|---|---|---|
| | N=20 | N=100 | Input | Output |
| GPT-4o-mini (AOT*) | 32.0 | 65.0 | 0.15 | 0.60 |
| DeepSeek-V3 (AOT*) | 67.0 | 86.0 | 0.56 | 1.68 |
| GPT-4o (AOT*) | 64.0 | 85.0 | 2.50 | 10.00 |
| Claude-4-Sonnet (AOT*) | 63.0 | 79.0 | 3.00 | 15.00 |
| Gemini-2.5 Pro (AOT*) | 66.0 | 84.0 | 1.25 | 10.00 |
| DeepSeek-V3 (LLM-S.P.) | 13.0 | 74.0 | 0.56 | 1.68 |
| GPT-4o (LLM-S.P.) | 9.0 | 72.0 | 2.50 | 10.00 |

The consistent 5-6× efficiency gap between AOT* and LLM-Syn-Planner across all models confirms that performance gains stem from our algorithmic framework, enabling practical cost-effective model deployment while maintaining superior efficiency.

### 4.7 FURTHER RESULTS AND VISUALIZATIONS

We provide comprehensive supplementary materials in the Appendices. Appendix B details dataset statistics, LLM configurations, and hyperparameter settings. The complete AOT* pseudocode, reaction validation details, baselines' descriptions, detailed comparisons with LLM-Syn-Planner, and detailed prompt usage are also included. Appendix C presents extended experimental results including performance comparisons across 11 LLM models, iteration efficiency analysis, additional difficulty-stratified performance breakdowns results, detailed cost-performance trade-offs results, along with additional ablation studies, hyperparameter sensitivity analyses, and visualization showcases of both success and failure synthesis trees cases. We provided LLMs usage statement at Appendix A, and discussions for limitations and future work at Appendix D.

## 5 CONCLUSION

In this work, we introduce AOT*, a novel framework that enhances the efficiency of multi-step retrosynthetic planning by integrating Large Language Models with AND-OR tree search. Our key innovation lies in atomically mapping LLM-generated synthesis pathways to AND-OR tree structures, preserving strategic coherence and enabling systematic intermediate reuse. This approach, combined with retrieval-augmented generation and systematic tree exploration, transforms the search process from iterative pathway optimization to structured exploration with pathway-level generation and achieves satisfying performance within constrained budgets. Extensive experiments demonstrate that AOT* achieves superior efficiency, requiring much fewer iterations than existing approaches to discover viable synthesis pathways while maintaining competitive solve rates across multiple synthesis benchmarks.

**Ethics Statement**   We confirm that this research complies with all applicable ethical guidelines and does not present any ethical issues.

**Reproducibility Statement**   To ensure reproducibility, we provide anonymized source code through the link in the abstract. Complete details regarding datasets, experimental settings, and implementation are documented in Appendix B.

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

# A USE OF LLMs

Large Language Models were used as assistive tools in the preparation of this manuscript. We employed LLMs for grammar checking, LaTeX formatting, improving the clarity of technical descriptions, and assisting with experimental code refactoring and implementation. The core scientific contributions and conclusions presented in this paper originate from the authors' work.

# B REPRODUCTIVITY

## B.1 EXPERIMENTAL SETUP

### B.1.1 DATASET STATISTICS

Table 6 summarizes the key characteristics that differentiate the datasets in terms of molecular complexity.

Table 6: Detailed statistics of benchmark datasets including molecular complexity metrics.

| Metric | USPTO-Easy | USPTO-190 | Pistachio Reachable | Pistachio Hard |
|---|---|---|---|---|
| Number of molecules | 200 | 190 | 150 | 100 |
| Avg. molecular weight | 382.1 | 458.6 | 446.2 | 467.4 |
| Avg. number of rings | 3.12 | 3.83 | 3.55 | 3.66 |
| Avg. chiral centers | 0.51 | 1.83 | 0.77 | 1.71 |
| Avg. SC score | 2.77 | 3.57 | 3.08 | 3.62 |

The statistics reveal that USPTO-Easy and Pistachio Reachable contains simpler molecules with lower molecular weight and SC scores, while USPTO-190 and Pistachio Hard feature more complex structures with higher chiral complexity, aligning with their intended difficulty levels.

### B.1.2 LLM MODELS

To evaluate the generalizability of our AOT* framework across different language model architectures, we tested multiple state-of-the-art LLM APIs including GPT-4o (gpt-4o-20250514) and GPT-4o-mini (Hurst et al., 2024), DeepSeek-V3 and DeepSeek-R1 (Liu et al., 2024; Guo et al., 2025), Claude-4-Sonnet (claude-sonnet-4-20250514) (Anthropic, 2025), Gemini-2.5 Pro (Comanici et al., 2025), Grok-4 (xAI, 2025), Qwen-3-MAX (Qwen-3-MAX-preview) (Team, 2025), and Llama-3.1-405B/Llama-3.1-70B (Grattafiori et al., 2024).

### B.1.3 HYPERPARAMETER SETTINGS

Our AOT* implementation uses the following hyperparameters: UCB exploration parameter $c = 0.5$, maximum search depth of 16. For LLM configuration, we set temperature T=0.7, maximum tokens of 4096, and use 3 RAG examples. The evaluation function weights availability at $\alpha = 0.4$. System-level parameters include 40 parallel threads for molecular planning searches until task completion. For DeepSeek-R1, we set maximum tokens to 32768 to accommodate its reasoning process and prevent output truncation (see Table 13 for output token statistics). All models were accessed through their respective commercial APIs with default parameters except for temperature and maximum tokens as specified.

### B.1.4 ALGORITHM IMPLEMENTATION

**AOT* pseudocode** Algorithm 1 presents the complete pseudocode for our AOT* framework.

---

**Algorithm 1** AOT*: AND-OR Tree Search with Generative Expansion.

---

**Require:** Target molecule $t$, building blocks $\mathcal{B}$, LLM generator $g$, database $\mathcal{D}$, max iterations $I_{\max}$, max depth $d_{\max}$
**Ensure:** Synthesis tree $\mathcal{T}^*$ or partial solution
 1: **Initialize:** $\mathcal{T} = (\mathcal{V}_{OR} = \{t\}, \mathcal{V}_{AND} = \emptyset, \mathcal{E} = \emptyset)$
 2: $\mathcal{L} \leftarrow \emptyset$ {Leaf AND nodes}
 3: $\mathcal{S}(t) \leftarrow \text{RetrieveSimilar}(t, \mathcal{D}, k)$ {Top-$k$ similar routes}
 4: $\mathcal{P}_t \leftarrow g(t, \mathcal{S}(t))$ {Generate initial pathways}
 5: $\mathcal{A}_{\text{init}} \leftarrow \Psi(\mathcal{P}_t, \mathcal{T})$ {Map pathways to tree}
 6: **for** $a \in \mathcal{A}_{\text{init}}$ **do**
 7:     $\bar{v}_a \leftarrow R(a) = \alpha \cdot f_{\text{avail}}(a) + (1 - \alpha) \cdot f_{\text{chem}}(a)$
 8:     $n_a \leftarrow 1$
 9:     $\mathcal{L} \leftarrow \mathcal{L} \cup \{a\}$ if $a$ has unsolved reactants
10: **end for**
11: $iter \leftarrow 0$
12: **while** $\neg\text{IsSolved}(t, \mathcal{T})$ **and** $iter < Iter_{\max}$ **do**
13:     **Selection:**
14:     $\mathcal{L}_{\text{expand}} \leftarrow \{a \in \mathcal{L} : d(a) < d_{\max} \wedge \exists v \in \text{Children}(a) : v \notin \mathcal{B}\}$
15:     **if** $|\mathcal{L}_{\text{expand}}| = 0$ **then**
16:         **break** {No expandable nodes}
17:     **end if**
18:     $a^* \leftarrow \arg\max_{a \in \mathcal{L}_{\text{expand}}} \text{UCB}(a)$ where
19:         $\text{UCB}(a) = \bar{v}_a + c\sqrt{\frac{\ln N_{\text{parent}}}{n_a}}$
20:     **Expansion:**
21:     $\mathcal{U} \leftarrow \{v \in \text{Children}(a^*) : v \notin \mathcal{B} \wedge \neg\text{IsSolved}(v)\}$
22:     $v^* \leftarrow \text{SelectTarget}(\mathcal{U})$ {Select least-explored molecule}
23:     $\mathcal{S}(v^*) \leftarrow \text{RetrieveSimilar}(v^*, \mathcal{D}, k)$
24:     $\mathcal{P}_{v^*} \leftarrow g(v^*, \mathcal{S}(v^*))$ {Generate pathways for $v^*$}
25:     $\mathcal{A}_{\text{new}} \leftarrow \Psi(\mathcal{P}_{v^*}, \mathcal{T})$ {Map to subtree}
26:     **Evaluation:**
27:     **for** $a \in \mathcal{A}_{\text{new}}$ **do**
28:         $r \leftarrow R(a) = \alpha \cdot f_{\text{avail}}(a) + (1 - \alpha) \cdot f_{\text{chem}}(a)$
29:         $\bar{v}_a \leftarrow r, n_a \leftarrow 1$
30:         $\mathcal{L} \leftarrow \mathcal{L} \cup \{a\}$ if $a$ has unsolved reactants
31:     **end for**
32:     **Backpropagation:**
33:     **for** $a \in \mathcal{A}_{\text{new}}$ **do**
34:         Propagate value $r$ to ancestors: $\forall a_p \in \text{Ancestors}(a)$:
35:             $\bar{v}_{a_p} \leftarrow \frac{n_{a_p} \cdot \bar{v}_{a_p} + r}{n_{a_p} + 1}$
36:             $n_{a_p} \leftarrow n_{a_p} + 1$
37:     **end for**
38:     $\text{UpdateSolvedStatus}(\mathcal{T}, \mathcal{B})$ {Propagate solved status}
39:     $\mathcal{L} \leftarrow \mathcal{L} \setminus \{a : \text{IsSolved}(a)\}$ {Remove solved nodes}
40:     $iter \leftarrow iter + 1$
41: **end while**
42: **if** $\text{IsSolved}(t, \mathcal{T})$ **then**
43:     **return** $\text{ExtractCompleteSolution}(\mathcal{T}, t)$
44: **else**
45:     **return** $\text{ExtractPartialSolution}(\mathcal{T}, t)$ {Return best partial tree}
46: **end if**

---

**Pathway-to-Tree Mapping** Algorithm 2 details the mapping procedure $\Psi$ that transforms LLM-generated pathways into AND-OR tree structures while maintaining consistency constraints.

**Subtree Pruning** Algorithm 3 describes the pruning procedure that removes solved subtrees from the active search space after molecules are resolved.

---

**Algorithm 2** Pathway-to-Tree Mapping $\Psi$.

---

**Require:** Pathway $p = \langle r_1, \ldots, r_n \rangle$, AND-OR tree $\mathcal{T}$, base depth $d$
**Ensure:** Set of new AND nodes $\mathcal{A}_{\text{new}}$
 1: $\mathcal{A}_{\text{new}} \leftarrow \emptyset$
 2: **for** $i = 1$ to $n$ **do**
 3:     Parse $r_i = (P_i \rightarrow \{R_{i,1}, \ldots, R_{i,k_i}\})$
 4:     $P_{\text{canon}} \leftarrow \text{Canonicalize}(P_i)$ {SMILES canonicalization}
 5:     **Find target OR node:**
 6:     **if** $P_{\text{canon}} \in \mathcal{V}_{OR}$ **then**
 7:         $v_{\text{product}} \leftarrow \mathcal{V}_{OR}[P_{\text{canon}}]$
 8:     **else**
 9:         **continue** {Skip orphaned steps}
10:     **end if**
11:     **if** IsSolved($v_{\text{product}}$) **then**
12:         **continue** {Skip solved molecules}
13:     **end if**
14:     **Create AND node:**
15:     $a_{\text{new}} \leftarrow \text{ANDNode}(r_i, v_{\text{product}}, d + i)$
16:     $\text{Children}(v_{\text{product}}) \leftarrow \text{Children}(v_{\text{product}}) \cup \{a_{\text{new}}\}$
17:     **Create/link reactant OR nodes:**
18:     **for** $j = 1$ to $k_i$ **do**
19:         $R_{\text{canon}} \leftarrow \text{Canonicalize}(R_{i,j})$
20:         **if** $R_{\text{canon}} \notin \mathcal{V}_{OR}$ **then**
21:             $v_{\text{reactant}} \leftarrow \text{ORNode}(R_{\text{canon}})$
22:             $\mathcal{V}_{OR} \leftarrow \mathcal{V}_{OR} \cup \{v_{\text{reactant}}\}$
23:             IsSolved($v_{\text{reactant}}$) $\leftarrow R_{\text{canon}} \in \mathcal{B}$
24:         **else**
25:             $v_{\text{reactant}} \leftarrow \mathcal{V}_{OR}[R_{\text{canon}}]$
26:         **end if**
27:         $\text{Children}(a_{\text{new}}) \leftarrow \text{Children}(a_{\text{new}}) \cup \{v_{\text{reactant}}\}$
28:         $\text{Parents}(v_{\text{reactant}}) \leftarrow \text{Parents}(v_{\text{reactant}}) \cup \{a_{\text{new}}\}$
29:     **end for**
30:     $\mathcal{V}_{AND} \leftarrow \mathcal{V}_{AND} \cup \{a_{\text{new}}\}$
31:     $\mathcal{A}_{\text{new}} \leftarrow \mathcal{A}_{\text{new}} \cup \{a_{\text{new}}\}$
32: **end for**
33: **return** $\mathcal{A}_{\text{new}}$

---

**RAG Database and Reaction Validation** Our retrieval-augmented generation utilizes a comprehensive reaction database constructed from USPTO training and validation sets (Wang et al., 2025). Table 7 summarizes the database statistics.

Table 7: RAG database statistics

| Property | Value |
|---|---|
| Total synthesis routes | 364,555 |
| Unique target molecules | 363,943 |
| Single-step routes | 192,710 (52.9%) |
| Two-step routes | 85,958 (23.6%) |
| Three-step routes | 43,592 (12.0%) |
| Routes with $\geq 4$ steps | 42,295 (11.6%) |

Besides, we followed the reaction validation method in (Wang et al., 2025), which employs a multi-level matching strategy: LLM-generated reactions are first searched for exact matches in the USPTO reaction database containing over 270k reaction templates; if no exact match is found, the top 100 most similar reactions are retrieved based on reaction fingerprint similarity and filtered by assessing chemical feasibility for the given product molecule, with the most similar valid reaction retained to

---

**Algorithm 3** Pruning Solved Subtrees.

---

**Require:** Set of newly solved molecules $\mathcal{M}_{\text{solved}}$, leaf nodes $\mathcal{L}$
**Ensure:** Updated leaf set $\mathcal{L}'$
 1: **function** PruneRecursive($a$):
 2:  **for** $v \in$ Children($a$) **do**
 3:    $\mathcal{U} \leftarrow \{a' \in$ Parents($v$) : $\neg$IsSolved($a'$)$\}$
 4:    **if** $|\mathcal{U}| = 0$ **then** {No unsolved parents}
 5:      **for** $a' \in$ Children($v$) **do**
 6:        $\mathcal{L} \leftarrow \mathcal{L} \setminus \{a'\}$
 7:        PruneRecursive($a'$) {Recursive cleanup}
 8:      **end for**
 9:    **end if**
10:  **end for**
11: **end function**
12:
13: **for** $m \in \mathcal{M}_{\text{solved}}$ **do**
14:   $v \leftarrow \mathcal{V}_{OR}[m]$
15:   **for** $a \in$ Children($v$) **do**
16:     $\mathcal{L} \leftarrow \mathcal{L} \setminus \{a\}$ {Remove from leaf set}
17:     PruneRecursive($a$)
18:   **end for**
19: **end for**
20: **return** $\mathcal{L}$

---

replace the LLM's original proposal; reactions without valid matches are labeled as non-existent. The method performs reaction mapping to ground the LLM generated routes against template set, effectively preventing hallucinated reactions by constraining outputs to verified chemical transformations. During tree expansion, generated pathways undergo three possible validation outcomes: (i) complete mapping success where all reactions match existing templates and the entire pathway is integrated into the tree structure; (ii) partial validation where only initial reaction steps successfully map to templates, with the valid portion incorporated while subsequent invalid steps are discarded; (iii) complete validation failure where no reactions match templates, causing the pathway expansion to be skipped without further processing. AND nodes $a \in \mathcal{V}_{AND}$ that repeatedly fail to produce valid pathways through the generative function $g$ are marked as non-expandable and excluded from future selection, ensuring the search focuses on productive regions of $\mathcal{T}$.

### B.2 BASELINE METHODS

**Graph2Edits** (Zhong et al., 2023) is a template-free graph generative model that directly edits molecular graphs to predict reactants from products. The method learns to systematically transform the target molecule's graph structure through a sequence of graph editing operations, including bond deletions, bond additions, and atom modifications. By treating retrosynthesis as a graph generation problem, Graph2Edits can handle diverse reaction types without relying on predefined templates, enabling it to generalize to novel reactions not seen during training.

**RootAligned** (Zhong et al., 2022) takes an alternative template-free approach by enforcing strict one-to-one correspondence between product and reactant SMILES representations. The method aligns both product and reactant molecules to a shared root atom, maintaining structural consistency throughout the retrosynthetic transformation. This alignment strategy ensures that the model learns meaningful chemical transformations while preserving the underlying molecular topology, leading to more interpretable and chemically valid predictions.

**LocalRetro** (Chen & Jung, 2021) adopts a template-based strategy that decomposes the retrosynthesis problem into two stages: local reaction center identification and global reactant completion. The method first predicts local templates describing atom and bond editing patterns at the reaction center, then employs a global attention mechanism to complete the full reactant structures by capturing non-local molecular effects. This hierarchical approach combines the interpretability of template-based methods with the flexibility to handle complex long-range dependencies in molecular structures.

These single-step models are integrated with two classical search algorithms. MCTS (Segler et al., 2018) performs Monte Carlo Tree Search to navigate the retrosynthesis space, iteratively building a search tree that balances exploration of new synthetic routes with exploitation of promising pathways. Retro* (Chen et al., 2020) performs best-first search on AND-OR trees where OR nodes represent molecules and AND nodes represent reactions, using neural networks to estimate node costs and prioritize the most promising pathways.

**DESP** (Yu et al., 2024) employs a bidirectional search strategy that simultaneously explores synthetic routes from both the target molecule (backward) and available starting materials (forward). The method uses neural networks to predict reactions in both directions and identifies viable synthesis plans when the forward and backward search frontiers meet, effectively reducing the search space by leveraging complementary information from both ends of the synthetic pathway.

**Tango*** guides retrosynthetic search from target molecules towards specified starting materials using the TANGO value function based on TANimoto Group Overlap. The method combines molecular similarity measures with retrosynthetic cost estimates to navigate the search space and identify synthesis pathways connecting the desired starting materials to target molecules.

Additionally, we compare against LLM-based approaches. **LLM (MCTS/Retro*)** (Wang et al., 2025) directly employs large language models as single-step reaction predictors within traditional search frameworks, using the LLM's chemical knowledge to propose reaction templates and predict feasible transformations at each step. **LLM-Syn-Planner** (Wang et al., 2025) also generates complete multi-step retrosynthetic routes using LLMs with retrieval augmentation, then iteratively refines them through evolutionary algorithms with mutation and selection operators. Both LLM-Syn-Planner and AOT* leverage LLMs for pathway-level generation with RAG; the key distinction lies in their search strategies—evolutionary optimization versus systematic AND-OR tree exploration with intermediate reuse. Here we further clarify AOT*'s architectural advantages by directly comparing with LLM-Syn-Planner (Wang et al., 2025), the current state-of-the-art in LLM-based retrosynthesis planning. Table 8 summarizes the key architectural differences between the two approaches.

Table 8: Architectural comparison between AOT* and LLM-Syn-Planner.

| Design Aspect | AOT* (Ours) | LLM-Syn-Planner |
|---|---|---|
| Generation Unit | Complete routes | Complete routes |
| Search Framework | AND-OR tree | Population-based EA |
| Route Integration | Tree mapping | Mutation/crossover |
| Exploration Strategy | UCB-guided expansion | Evolutionary operators |
| Intermediate Reuse | Tree-wide sharing | No reuse |
| Memory Structure | Search tree | Population pool |

In Table 1, the results for DESP and Tango* are obtained from their original papers (Yu et al., 2024; Jončev et al., 2025), while all other baseline results (excluding AOT*) are from (Wang et al., 2025); all remaining results throughout the paper are from our own implementation.

### B.3 PROMPTS

We maintain identical prompt configurations and structure to ensure fair comparison with LLM-Syn-Planner (Wang et al., 2025). The prompts consist of modular components that guide LLMs toward chemically valid retrosynthesis routes. Each component can be ablated independently to assess its contribution to search performance.

**Role Information Component** The role definition establishes chemistry expert context for the LLM (Figure 4).

**Task Description Component** The task description defines retrosynthesis fundamentals and iterative process (Figure 5). When ablated, it reduces to: "Propose a retrosynthesis route for the target molecule."

**Role Information Component**

```
You are a professional chemist specializing in synthesis analysis.
```

Figure 4: Role information component.

**Task Description Component**

```
Your task is to propose a retrosynthesis route for a target molecule
provided in SMILES format.

Definition:
A retrosynthesis route is a sequence of backward reactions that starts
from the target molecules and ends with commercially purchasable
building blocks.

Key concepts:
- Molecule set: The working set of molecules at any given step.
  Initially, it contains only the target molecule.
- Commercially purchasable: Molecules that can be directly bought
  from suppliers (permitted building blocks).
- Non-purchasable: Molecules that must be further decomposed via
  retrosynthesis steps.
- Reaction source: All reactions must be derived from the USPTO
  dataset, and stereochemistry (e.g., E/Z isomers, chiral centers)
  must be preserved.

Process:
1. Initialization: Start with the molecule set = [target molecule].
2. Iteration:
   - Select one non-purchasable molecule from the molecule set
     (the product).
   - Apply a valid backward reaction from the USPTO dataset to
     decompose it into reactants.
   - Remove the product molecule from the set.
   - Add the reactants to the set.
3. Termination: Continue until all molecules in the set are
   commercially purchasable.
```

Figure 5: Task description component.

**RAG Integration Component**   The RAG component retrieves similar synthesis routes to guide generation (Figure 6).

**RAG Integration Component**

```
My target molecule is: {target_smiles}

To assist you with the format, example retrosynthesis routes are
provided:
{examples}

Please propose {rag\_examples} different retrosynthesis routes
for my target molecule.
```

Figure 6: RAG integration with retrieved examples.

**Planning Requirement Component**    The planning component requires strategic analysis before route generation (Figure 7).

---

**Planning Requirement Component**

```
analyze the target molecule and make a retrosynthesis plan in the
<PLAN></PLAN> before proposing the route.

<PLAN>: Analyze the target molecule and plan for each step in the
route. </PLAN>
```

Figure 7: Planning requirement component.

**Explanation Requirement Component**    The explanation component requires justification of the proposed plan (Figure 8).

---

**Explanation Requirement Component**

```
After making the plan, you should explain the plan in the
<EXPLANATION></EXPLANATION>.

<EXPLANATION>: Explain the plan. </EXPLANATION>
```

Figure 8: Explanation requirement component.

**Structured Output Format with Rational Field**    The output format defines the route structure with optional rational field (Figure 9).

---

**Structured Output Format**

```
The route should be a list of steps wrapped in <ROUTE></ROUTE>.
Each step in the list should be a dictionary.
At the first step, the molecule set should be the target molecules
set given by the user. Here is an example:

<ROUTE>
[
    {
        'Molecule set': "[Target Molecule]",
        'Rational': "Step analysis",  # Ablated with no_rational
        'Product': "[Product molecule]",
        'Reaction': "[Reaction template]",  # Ablated with no_reaction
        'Reactants': "[Reactant1, Reactant2]",
        'Updated molecule set': "[Reactant1, Reactant2]"
    }
]
</ROUTE>
```

Figure 9: Structured output format.

**Detailed Requirements Section**    The detailed requirements provide field-by-field specifications, dynamically built based on ablation settings (Figure 10).

```
Detailed Requirements

  1. The 'Molecule set' contains molecules we need to synthesize at
     this stage. In the first step, it should be the target molecule.
     In the following steps, it should be the 'Updated molecule set'
     from the previous step.

  2. The 'Rational' part in each step should be your analysis for
     synthesis planning in this step. It should be in the string
     format wrapped with ''

  3. 'Product' is the molecule we plan to synthesize in this step.
     It should be from the 'Molecule set'. The molecule should be a
     molecule from the 'Molecule set' in a list. The molecule smiles
     should be wrapped with ''.

  4. 'Reaction' is a backward reaction which can decompose the product
     molecule into its reactants. The reaction should be in a list.
     All the molecules in the reaction template should be in SMILES
     format. [Only if no_reaction is False; simplified format
     available via simple_reaction_format]

  5. 'Reactants' are the reactants of the reaction. It should be in
     a list. The molecule smiles should be wrapped with ''.

  6. The 'Updated molecule set' should be molecules we need to purchase
     or synthesize after taking this reaction. To get the 'Updated
     molecule set', you need to remove the product molecule from the
     'Molecule set' and then add the reactants in this step into it.
     In the last step, all the molecules in the 'Updated molecule set'
     should be purchasable.

  7. In the <PLAN>, you should analyze the target molecule and plan
     for the whole route. [Only if no_plan is False]

  8. In the <EXPLANATION>, you should analyze the plan.
     [Only if no_explanation is False]
```

Figure 10: Detailed requirements section dynamically constructed based on ablation flags. Requirements are numbered sequentially with conditional inclusion.

## C  EXTENDED EXPERIMENTAL RESULTS

Results reported in this section use 100 iterations (N = 100) as the default search budget unless otherwise specified.

### C.1  EXTENDED LLM MODEL COMPARISON

Table 9 presents AOT* performance with 11 different LLMs on Pistachio Hard and USPTO-190 datasets.

#### C.1.1  MAIN PERFORMANCE COMPARISON

Table 9 presents solve rates across different search budgets for various LLM architectures. DeepSeek-R1 achieves the highest performance, with 89.0% solve rate on Pistachio Hard and 90.5% on USPTO-190 at N=100 iterations. A cluster of models including GPT-4o, GPT-5, DeepSeek-V3, Gemini-2.5 Pro, and Grok-4 achieve similar performance ranging from 83-86% on both datasets at N=100. Claude-4-Sonnet and Llama-3.1-405B perform moderately lower at 74-79%, while smaller models show significant performance gaps: GPT-4o-mini achieves 65.0% and 54.2%, and Llama-3.1-70B reaches only 73.0% and 63.2% on the two benchmarks respectively. Increasing the search

budget from N=100 to N=300 provides substantial improvements for most models. However, further expansion to N=500 yields diminishing returns, typically adding only 2-4% additional solve rate. This saturation pattern is consistent across model scales, with most architectures reaching their performance ceiling around N=300. The results indicate that AOT*'s algorithmic framework maintains effectiveness across diverse LLM models, though absolute performance may correlate with model capability.

Table 9: Comparison of solve rates (%) across different LLM architectures on challenging benchmarks. Best results are **bolded** and top-3 are underlined.

| Model | Pistachio Hard | | | USPTO-190 | | |
|---|---|---|---|---|---|---|
| | N=100 | N=300 | N=500 | N=100 | N=300 | N=500 |
| GPT-4o | 85.0 | 87.0 | 93.0 | 82.1 | 92.6 | 93.1 |
| GPT-4o-mini | 65.0 | 68.0 | 72.0 | 54.2 | 67.4 | 71.6 |
| GPT-5 | 86.0 | 88.0 | 93.0 | 84.7 | 90.5 | 92.1 |
| DeepSeek-V3 | 86.0 | 89.0 | 93.0 | 86.3 | 93.1 | 93.7 |
| DeepSeek-R1 | **89.0** | **93.0** | **94.0** | **90.5** | **94.2** | **95.3** |
| Claude-4-Sonnet | 79.0 | 81.0 | 83.0 | 74.7 | 84.2 | 86.8 |
| Gemini-2.5 Pro | 84.0 | 86.0 | 89.0 | 78.4 | 86.3 | 88.9 |
| Grok-4 | 85.0 | 87.0 | 91.0 | 83.2 | 88.4 | 91.6 |
| Qwen-3-MAX | 83.0 | 86.0 | 92.0 | 80.0 | 87.9 | 91.1 |
| Llama-3.1-405B | 79.0 | 81.0 | 83.0 | 74.7 | 85.3 | 87.9 |
| Llama-3.1-70B | 73.0 | 74.0 | 75.0 | 63.2 | 75.8 | 78.9 |

### C.1.2 ITERATION EFFICIENCY ANALYSIS

Table 10 shows solve rates at different iteration thresholds (20, 40, 60, 80, 100) for each model. DeepSeek-R1 demonstrates the highest efficiency, achieving 76.0% solve rate within 20 iterations on Pistachio Hard and 67.9% on USPTO-190. GPT-5 and DeepSeek-V3 follow closely with 71.0% and 67.0% respectively on Pistachio Hard at 20 iterations. In contrast, smaller models exhibit significantly lower early-stage efficiency: GPT-4o-mini reaches only 32.0% on Pistachio Hard and 24.7% on USPTO-190 at 20 iterations, while Llama-3.1-70B achieves 51.0% and 31.6% respectively. The efficiency gap between models narrows as iterations increase. At 40 iterations, most full-scale models achieve 73-82% solve rates on Pistachio Hard, while GPT-4o-mini and Llama-3.1-70B remain at 45.0% and 60.0%. By 60 iterations, the leading models approach their performance plateaus, with DeepSeek-R1 at 85.0% and GPT-5 at 82.0% on Pistachio Hard. GPT-4o-mini requires approximately 60 iterations to reach solve rates that other models achieve at 20 iterations, indicating a 3× efficiency difference. On USPTO-190, DeepSeek-R1 maintains its efficiency advantage, reaching 80.5% at 40 iterations compared to other models. Most models show minimal improvement beyond 80 iterations, with solve rates increasing by only 2-4% from iteration 80 to 100, suggesting that additional iterations provide limited benefit regardless of model architecture.

### C.1.3 DIFFICULTY-STRATIFIED PERFORMANCE

Tables 11 and 12 break down model performance by SC score quartiles (Q1: simplest, Q4: most complex). All models exhibit consistent performance degradation as molecular complexity increases, with solve rates typically dropping 20-30% from Q1 to Q4. Most full-scale models achieve near-perfect performance on simple molecules (Q1: 92-100%), while their performance on the most complex quartile varies significantly based on model capability. DeepSeek-R1 maintains the strongest performance across all complexity levels, achieving 80.0% solve rate on Pistachio Hard Q4 and 83.0% on USPTO-190 Q4. This represents only a 20% drop from its Q1 performance, compared to larger degradations in other models. Smaller models show particular vulnerability to increasing complexity: GPT-4o-mini drops from 84.0% to 52.0% on Pistachio Hard and from 72.9% to 38.3% on USPTO-190, while Llama-3.1-70B falls to 60.0% and 46.8% respectively on Q4 molecules.

Iteration requirements also scale with molecular complexity. Simple molecules (Q1) typically require fewer than 20 iterations across all models, while complex molecules (Q4) demand 30-70 iterations depending on model capability. This scaling effect is more pronounced in weaker models:

Table 10: Comparison of solve rates (%) at different iteration thresholds across LLM architectures. Best results are **bolded** and top-3 are underlined.

| Model | Pistachio Hard | | | | | USPTO-190 | | | | |
|---|---|---|---|---|---|---|---|---|---|---|
| | 20 | 40 | 60 | 80 | 100 | 20 | 40 | 60 | 80 | 100 |
| GPT-4o | 64.0 | 76.0 | 79.0 | 81.0 | 85.0 | 55.7 | 69.5 | 78.4 | 80.5 | 82.1 |
| GPT-4o-mini | 32.0 | 45.0 | 55.0 | 62.0 | 65.0 | 24.7 | 34.7 | 41.6 | 47.9 | 54.2 |
| GPT-5 | 71.0 | 78.0 | 82.0 | 83.0 | 85.0 | 57.9 | 73.7 | 80.0 | 82.6 | 84.7 |
| DeepSeek-V3 | 67.0 | 78.0 | 81.0 | 83.0 | 86.0 | 56.3 | 72.1 | 81.6 | 85.3 | 86.3 |
| DeepSeek-R1 | **76.0** | **82.0** | **85.0** | **87.0** | **89.0** | **67.9** | **80.5** | **85.8** | **88.9** | **90.5** |
| Claude-4-Sonnet | 63.0 | 67.0 | 70.0 | 75.0 | 79.0 | 41.6 | 55.8 | 64.7 | 70.0 | 74.7 |
| Gemini-2.5 Pro | 66.0 | 78.0 | 81.0 | 83.0 | 84.0 | 46.8 | 62.6 | 70.5 | 74.7 | 78.4 |
| Grok-4 | 65.0 | 76.0 | 83.0 | 84.0 | 85.0 | 52.6 | 68.9 | 75.8 | 80.0 | 83.2 |
| Qwen-3-MAX | 65.0 | 73.0 | 77.0 | 80.0 | 83.0 | 47.9 | 61.6 | 70.5 | 75.8 | 80.0 |
| Llama-3.1-405B | 58.0 | 69.0 | 76.0 | 79.0 | 79.0 | 38.9 | 51.6 | 62.6 | 68.9 | 74.7 |
| Llama-3.1-70B | 51.0 | 60.0 | 71.0 | 72.0 | 73.0 | 31.6 | 42.6 | 52.6 | 57.9 | 63.2 |

GPT-4o-mini requires 64.3 iterations for Pistachio Hard Q4 compared to DeepSeek-R1's 25.8 iterations. The iteration efficiency gap between models widens substantially as complexity increases, reinforcing that model capability becomes increasingly critical for challenging synthesis problems.

Table 11: Performance breakdown by SC score quartiles for Pistachio Hard dataset. Best results are **bolded** and top-3 are underlined.

| Model | Solve Rate (%) | | | | Avg SR | Iterations | | | | Avg Iter. |
|---|---|---|---|---|---|---|---|---|---|---|
| | Q1 | Q2 | Q3 | Q4 | | Q1 | Q2 | Q3 | Q4 | |
| GPT-4o | **100.0** | 88.0 | 80.0 | 72.0 | 85.0 | 5.8 | 18.3 | 26.0 | 39.0 | 22.3 |
| GPT-4o-mini | 84.0 | 68.0 | 56.0 | 52.0 | 65.0 | 22.5 | 51.2 | 65.8 | 64.3 | 50.9 |
| GPT-5 | **100.0** | 88.0 | 80.0 | 72.0 | 85.0 | 4.4 | 16.8 | 21.4 | 34.0 | 19.1 |
| DeepSeek-V3 | **100.0** | 88.0 | 80.0 | 76.0 | 86.0 | 5.8 | 13.9 | 28.7 | 32.9 | 20.3 |
| DeepSeek-R1 | **100.0** | **92.0** | **84.0** | **80.0** | **89.0** | **3.8** | **12.5** | **18.6** | **25.8** | **15.2** |
| Claude-4-Sonnet | 96.0 | 88.0 | 68.0 | 64.0 | 79.0 | 4.9 | 22.4 | 39.5 | 52.3 | 29.8 |
| Gemini-2.5 Pro | **100.0** | **92.0** | 76.0 | 68.0 | 84.0 | 4.7 | 20.1 | 31.5 | 32.7 | 22.3 |
| Grok-4 | **100.0** | **92.0** | 80.0 | 68.0 | 85.0 | 6.3 | 13.0 | 30.0 | 35.1 | 21.1 |
| Qwen-MAX | 92.0 | 88.0 | **84.0** | 68.0 | 83.0 | 10.8 | 15.4 | 26.4 | 37.8 | 22.6 |
| Llama-3.1-405B | 92.0 | **92.0** | 76.0 | 56.0 | 79.0 | 8.8 | 20.4 | 39.6 | 51.0 | 29.9 |
| Llama-3.1-70B | 96.0 | 84.0 | 44.0 | 60.0 | 71.0 | 13.9 | 25.0 | 60.8 | 53.9 | 38.4 |

Table 12: Performance breakdown by SC score quartiles for USPTO-190 dataset. Best results are **bolded** and top-3 are underlined.

| Model | Solve Rate (%) | | | | Avg SR | Iterations | | | | Avg Iter. |
|---|---|---|---|---|---|---|---|---|---|---|
| | Q1 | Q2 | Q3 | Q4 | | Q1 | Q2 | Q3 | Q4 | |
| GPT-4o | 97.9 | 89.4 | 77.1 | 63.8 | 82.1 | 16.2 | 27.3 | 39.9 | 45.5 | 32.2 |
| GPT-4o-mini | 72.9 | 59.6 | 45.8 | 38.3 | 54.2 | 34.8 | 45.2 | 58.6 | 67.3 | 51.5 |
| GPT-5 | 97.9 | 87.2 | 79.2 | 76.6 | 85.3 | 14.7 | 24.1 | 35.8 | 41.2 | 28.9 |
| DeepSeek-V3 | **100.0** | 85.1 | 81.2 | 78.7 | 86.3 | 18.9 | 27.7 | 36.8 | 40.3 | 29.9 |
| DeepSeek-R1 | **100.0** | **91.5** | **87.5** | **83.0** | **90.5** | **11.3** | **19.8** | **26.4** | **31.7** | **22.3** |
| Claude-4-Sonnet | 91.7 | 80.9 | 70.8 | 61.7 | 76.3 | 21.4 | 33.7 | 46.9 | 56.2 | 39.5 |
| Gemini-2.5 Pro | 93.8 | 85.1 | 75.0 | 63.8 | 79.5 | 19.8 | 31.4 | 43.7 | 50.6 | 36.4 |
| Grok-4 | 97.9 | 83.0 | 77.1 | 74.5 | 83.2 | 17.6 | 29.3 | 40.8 | 47.1 | 33.7 |
| Qwen-3-MAX | 95.8 | 83.0 | 75.0 | 72.3 | 81.6 | 24.3 | 36.8 | 48.9 | 58.4 | 42.1 |
| Llama-3.1-405B | 81.2 | 76.6 | 79.2 | 61.7 | 74.7 | 32.0 | 39.5 | 47.1 | 54.8 | 43.3 |
| Llama-3.1-70B | 79.2 | 68.1 | 58.3 | 46.8 | 63.2 | 36.5 | 49.7 | 62.1 | 69.8 | 54.5 |

### C.1.4 COST-PERFORMANCE ANALYSIS

Table 13 compares API costs and performance across models. DeepSeek-V3 offers the best value at $0.56/$1.68 per million tokens (input/output) with 86% solve rate, while GPT-4o-mini is cheapest ($0.15/$0.60) but achieves only 65% solve rate. DeepSeek-R1 matches DeepSeek-V3's pricing but generates 10× more output tokens due to its reasoning traces. Among premium models ($2.50+ per million input tokens), performance differences are minimal (79-85% solve rate). These results demonstrate that DeepSeek-V3 provides the optimal cost-performance balance for the testing experiments, achieving competitive performance without the substantial token overhead from thinking processes or the premium pricing of other models.

Table 13: Cost-performance trade-offs across different LLM architectures on benchmark datasets. Best results are **bolded** and top-3 are underlined. Green indicates low cost/tokens, red indicates high cost/tokens.

| Model | Token Cost ($/1M) | | Pistachio Hard | | | USPTO-190 | | |
| | Input | Output | SR (%) | Avg Iter. | Avg Output | SR (%) | Avg Iter. | Avg Output |
|---|---|---|---|---|---|---|---|---|
| GPT-4o-mini | 0.15 | 0.60 | 65.0 | 50.9 | 1,078 | 54.2 | 51.5 | 1,039 |
| DeepSeek-V3 | 0.56 | 1.68 | 86.0 | 20.3 | 1,221 | 86.3 | 29.9 | 1,611 |
| DeepSeek-R1 | 0.56 | 1.68 | **89.0** | **15.2** | 12,109 | **90.5** | **22.3** | 12,298 |
| GPT-5 | 1.25 | 10.00 | 85.0 | 19.1 | 1,862 | 84.7 | 28.9 | 1,502 |
| Gemini-2.5 Pro | 1.25 | 10.00 | 84.0 | 22.3 | 2,689 | 78.4 | 36.4 | 2,735 |
| Qwen-3-MAX | 1.20 | 6.00 | 83.0 | 22.6 | 2,462 | 80.0 | 42.1 | 2,051 |
| GPT-4o | 2.50 | 10.00 | 85.0 | 22.3 | 1,437 | 82.1 | 32.2 | 1,343 |
| Claude-4-Sonnet | 3.00 | 15.00 | 79.0 | 29.8 | 1,616 | 74.7 | 39.5 | 1,702 |
| Grok-4 | 3.00 | 15.00 | 85.0 | 21.1 | 2,949 | 83.2 | 33.7 | 2,184 |

## C.2 ADDITIONAL COMPONENT ANALYSIS RESULTS

We provide additional experimental results on component analysis in this section.

### C.2.1 FURTHER PROMPT ABLATION RESULTS

Figure 11 extends the prompt ablation analysis to the USPTO-Easy and Pistachio Reachable datasets, complementing the results from the more challenging benchmarks presented in the main text. On these simpler datasets, all configurations achieve high solve rates (>90%) by 100 iterations, but RAG removal still causes the most substantial early-stage degradation, with approximately 10-15% lower solve rates at 20 iterations. The performance gaps between ablated configurations narrow more rapidly compared to challenging datasets, with most differences becoming negligible beyond 60 iterations, suggesting that prompt components primarily accelerate convergence rather than determine ultimate performance ceilings on simpler synthesis problems.

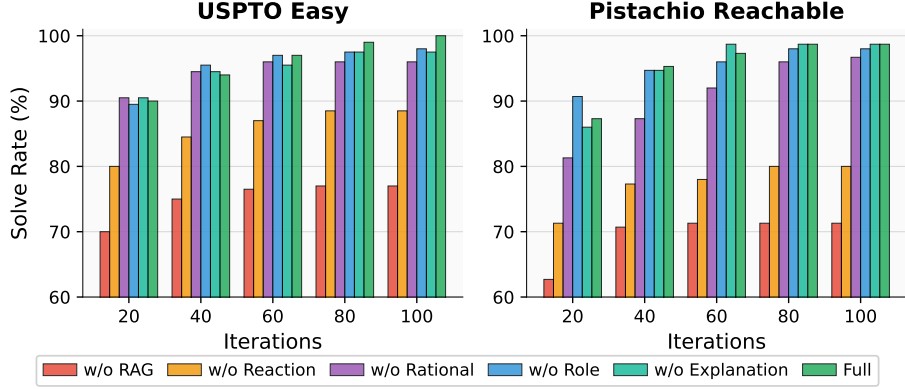

Figure 11: Impact of prompt components on solve rates for USPTO-Easy and Pistachio Reachable, N = 100.

**Difficulty-Stratified Ablation Analysis.** Tables 14, 15, 16, and 17 show how prompt ablations affect molecules of different complexities. RAG retrieval is critical across all difficulty levels—removing it drops Q4 performance by 32% on USPTO-190 and 28% on Pistachio Hard. Simple molecules (Q1) maintain high solve rates even without RAG (83-92%), while complex molecules (Q4) suffer dramatically without it (47-57%). Other components show minimal impact.

Table 14: Prompt ablation performance by SC score quartiles on USPTO-190.

| Configuration | Solve Rate (%) | | | | Avg. SR | Iterations | | | | Avg Iter. |
|---|---|---|---|---|---|---|---|---|---|---|
| | Q1 | Q2 | Q3 | Q4 | | Q1 | Q2 | Q3 | Q4 | |
| Full Prompt | 100.0 | 85.1 | 81.2 | 78.7 | 86.3 | 18.9 | 26.5 | 35.5 | 38.5 | 29.9 |
| No RAG | 83.3 | 46.8 | 45.8 | 46.8 | 55.8 | 19.8 | 29.0 | 36.4 | 29.9 | 28.8 |
| No Explanation | 97.9 | 83.0 | 79.2 | 76.6 | 84.2 | 23.1 | 30.5 | 35.2 | 39.0 | 31.9 |
| No Rational | 100.0 | 85.1 | 79.2 | 78.7 | 85.8 | 16.1 | 24.4 | 28.2 | 34.9 | 25.9 |
| No Role Info | 100.0 | 85.1 | 79.2 | 76.6 | 85.3 | 16.6 | 28.7 | 35.4 | 48.5 | 32.3 |
| No Reaction | 97.9 | 78.7 | 68.8 | 55.3 | 75.3 | 17.6 | 25.0 | 38.2 | 35.4 | 29.1 |

Table 15: Prompt ablation performance by SC score quartiles on Pistachio Hard.

| Configuration | Solve Rate (%) | | | | Avg. SR | Iterations | | | | Avg Iter. |
|---|---|---|---|---|---|---|---|---|---|---|
| | Q1 | Q2 | Q3 | Q4 | | Q1 | Q2 | Q3 | Q4 | |
| Full Prompt | 100.0 | 88.0 | 80.0 | 76.0 | 86.0 | 5.8 | 13.9 | 28.7 | 32.9 | 20.3 |
| No RAG | 84.0 | 56.0 | 52.0 | 48.0 | 60.0 | 15.5 | 26.7 | 26.4 | 23.2 | 23.0 |
| No Explanation | 88.0 | 88.0 | 76.0 | 80.0 | 83.0 | 4.3 | 17.9 | 38.4 | 22.9 | 20.9 |
| No Rational | 92.0 | 88.0 | 76.0 | 76.0 | 83.0 | 7.2 | 13.9 | 27.4 | 32.8 | 20.3 |
| No Role Info | 92.0 | 88.0 | 88.0 | 68.0 | 84.0 | 3.5 | 17.2 | 22.1 | 40.6 | 20.9 |
| No Reaction | 88.0 | 92.0 | 72.0 | 68.0 | 80.0 | 4.3 | 13.3 | 29.6 | 33.9 | 20.3 |

Table 16: Prompt ablation performance by SC score quartiles on Pistachio Reachable.

| Configuration | Solve Rate (%) | | | | Avg. SR | Iterations | | | | Avg Iter. |
|---|---|---|---|---|---|---|---|---|---|---|
| | Q1 | Q2 | Q3 | Q4 | | Q1 | Q2 | Q3 | Q4 | |
| Full Prompt | 100.0 | 100.0 | 97.3 | 97.4 | 98.7 | 4.8 | 9.5 | 9.7 | 12.1 | 9.0 |
| No RAG | 92.1 | 76.3 | 59.5 | 56.8 | 71.3 | 6.9 | 9.1 | 10.8 | 10.6 | 9.4 |
| No Explanation | 97.4 | 100.0 | 100.0 | 97.3 | 98.7 | 7.1 | 6.1 | 8.3 | 14.7 | 9.1 |
| No Rational | 100.0 | 97.4 | 97.3 | 91.9 | 96.7 | 5.7 | 9.8 | 14.4 | 28.6 | 14.6 |
| No Role Info | 100.0 | 100.0 | 100.0 | 91.9 | 98.0 | 4.9 | 9.2 | 6.4 | 18.1 | 9.6 |
| No Reaction | 84.2 | 81.6 | 89.2 | 64.9 | 80.0 | 5.8 | 7.9 | 13.5 | 16.8 | 11.0 |

**Cost of Prompt Components.** Table 18 shows the token-performance trade-off for each prompt component. Removing RAG reduces input tokens by approximately one-third but causes the largest performance degradation, dropping solve rates by over 25%. Role information also contributes substantially to token count (27% reduction when removed) with moderate performance impact. Minor components like reaction and rationale fields account for less than 5% of tokens each and show minimal effect on performance. The analysis reveals that token efficiency cannot be achieved through simple prompt reduction, as the most token-intensive components are also the most critical for maintaining search effectiveness.

## C.3   IMPACT OF RAG SAMPLE SIZE

Figure 12 illustrates the relationship between the number of RAG examples and solve rates for the USPTO-Easy and Pistachio Reachable datasets, demonstrating that performance gains plateau after 3-5 examples even for these simpler benchmarks.

Table 17: Prompt ablation performance by SC score quartiles on USPTO Easy.

| Configuration | Solve Rate (%) | | | | Avg. SR | Iterations | | | | Avg Iter. |
| | Q1 | Q2 | Q3 | Q4 | | Q1 | Q2 | Q3 | Q4 | |
|---|---|---|---|---|---|---|---|---|---|---|
| Full Prompt | 100.0 | 100.0 | 100.0 | 100.0 | 100.0 | 2.8 | 9.1 | 10.3 | 15.7 | 9.5 |
| No RAG | 92.0 | 80.0 | 80.0 | 56.0 | 77.0 | 2.9 | 9.3 | 8.8 | 9.9 | 7.7 |
| No Explanation | 100.0 | 96.0 | 98.0 | 96.0 | 97.5 | 1.8 | 11.0 | 6.5 | 13.1 | 8.1 |
| No Rational | 98.0 | 96.0 | 100.0 | 90.0 | 96.0 | 2.8 | 9.4 | 6.5 | 14.1 | 8.2 |
| No Role Info | 100.0 | 96.0 | 100.0 | 96.0 | 98.0 | 1.6 | 9.3 | 6.7 | 14.4 | 8.0 |
| No Reaction | 96.0 | 84.0 | 88.0 | 86.0 | 88.5 | 1.0 | 5.2 | 9.5 | 17.5 | 8.3 |

Table 18: Comprehensive ablation study results with token statistics and performance metrics.

| Configuration | Input Tokens | | Token | Performance | |
| | Mean | Std | Reduction (%) | Avg Iter. | SR (%) |
|---|---|---|---|---|---|
| No RAG | 995 | 19 | -32.7 | 22.4 | 60.0 |
| No Role Info | 1078 | 95 | -27.1 | 14.1 | 84.0 |
| No Explanation | 1328 | 245 | -10.1 | 14.1 | 83.0 |
| No Rational | 1428 | 305 | -3.4 | 15.3 | 83.0 |
| No Reaction | 1448 | 315 | -2.0 | 13.8 | 80.0 |
| Full Prompt | 1478 | 328 | - | 14.4 | 86.0 |

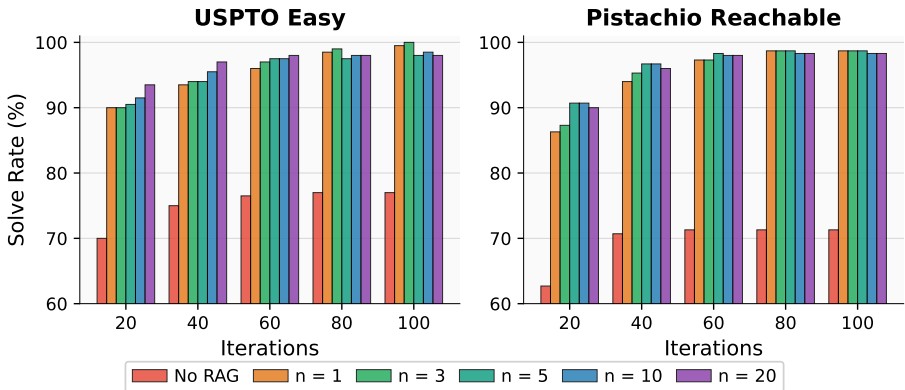

Figure 12: Impact of RAG sample number (n) on solve rates for USPTO-Easy and Pistachio Reachable, N = 100.

**Cost of RAG Samples.** Table 19 shows the diminishing returns of increasing RAG samples on Pistachio hard. Using 3 examples achieves 86% solve rate with 1,478 tokens, while 20 examples only improves performance by 2% but increases token usage by 177%. The sweet spot is 3-5 examples—beyond this, token costs grow exponentially with negligible performance gains.

## C.4 MOLECULAR WEIGHT-STRATIFIED ANALYSIS

Table 20 shows how performance degrades with increasing molecular weight. We divide each dataset into quartiles based on molecular weight distribution, where Q1 represents the smallest molecules and Q4 the largest. Larger molecules (Q4) consistently require more iterations and achieve lower solve rates across all datasets. The effect is most pronounced on challenging benchmarks—Pistachio Hard drops from 100% (Q1) to 76% (Q4).

## C.5 MOLECULAR WEIGHT-STRATIFIED ANALYSIS

Table 20 presents molecular weight statistics and corresponding performance metrics across all datasets. Performance consistently degrades with increasing molecular weight, with Q4 molecules

Table 19: RAG sample size impact on token usage and performance metrics, Pistachio Hard.

| RAG Samples | Input Tokens | | | | Token | Performance | |
| | Mean | Std | Min | Max | Change (%) | Avg Iter. | SR (%) |
|---|---|---|---|---|---|---|---|
| 0 (No RAG) | 995 | 19 | 965 | 1077 | -32.7 | 22.4 | 60.0 |
| 1 | 1172 | 135 | 1026 | 1718 | -20.7 | 15.0 | 82.0 |
| 3 | 1478 | 328 | 1104 | 3063 | 0.0 | 14.4 | 86.0 |
| 5 | 1780 | 517 | 1195 | 4318 | +20.4 | 15.7 | 86.0 |
| 10 | 2566 | 1024 | 1366 | 8492 | +73.6 | 12.4 | 86.0 |
| 20 | 4091 | 2038 | 2035 | 17223 | +176.7 | 12.5 | 88.0 |

requiring significantly more iterations and achieving lower solve rates compared to Q1. This degradation is particularly severe in challenging benchmarks, where Pistachio Hard's solve rate decreases by 24% from the smallest (Q1: 100%) to largest (Q4: 76%) molecules.

Table 20: Molecular weight (MW) quartile statistics and performance breakdown across datasets.

| Dataset | MW Average (g/mol) | | | | MW Range (g/mol) | | | |
| | Q1 | Q2 | Q3 | Q4 | Q1 | Q2 | Q3 | Q4 |
|---|---|---|---|---|---|---|---|---|
| Pistachio Hard | 267.6 | 395.8 | 501.3 | 702.7 | 163-354 | 354-448 | 448-561 | 561-1171 |
| USPTO-190 | 288.1 | 379.3 | 465.1 | 698.3 | 181-346 | 346-417 | 417-519 | 519-954 |
| USPTO-Easy | 246.0 | 348.1 | 416.2 | 516.6 | 182-299 | 299-388 | 388-447 | 447-686 |
| Pistachio Reachable | 267.3 | 397.2 | 484.2 | 634.2 | 127-342 | 342-439 | 439-533 | 533-1307 |

| Dataset | Solve Rate (%) | | | | Iterations | | | |
| | Q1 | Q2 | Q3 | Q4 | Q1 | Q2 | Q3 | Q4 |
|---|---|---|---|---|---|---|---|---|
| Pistachio Hard | 100.0 | 88.0 | 84.0 | 80.0 | 9.72 | 12.16 | 24.96 | 34.44 |
| USPTO-190 | 93.8 | 89.4 | 87.2 | 75.0 | 30.79 | 26.38 | 22.74 | 39.04 |
| USPTO-Easy | 100.0 | 100.0 | 100.0 | 100.0 | 6.54 | 6.90 | 7.20 | 16.00 |
| Pistachio Reachable | 100.0 | 100.0 | 97.3 | 97.4 | 7.76 | 9.62 | 8.68 | 10.08 |

## C.6 HYPERPARAMETER SENSITIVITY ANALYSIS

We analyze the sensitivity of AOT* to key hyperparameters on the Pistachio Hard dataset. All experiments use N=100 iterations with results stratified by molecular complexity (SC score quartiles).

### C.6.1 LLM GENERATION PARAMETERS

Table 21 shows the impact of LLM temperature on route generation quality. Temperature T=0.7 achieves optimal performance, balancing exploration and exploitation. Lower temperatures (T=0.1) reduce diversity, causing poor performance on complex molecules (Q4: 60%), while higher temperatures (T≥0.9) generate inconsistent routes despite maintaining reasonable solve rates.

Table 21: Temperature parameter impact on solve rates and iterations.

| T | Solve Rate (%) | | | | Avg SR | Iterations | | | | Avg Iter. |
| | Q1 | Q2 | Q3 | Q4 | | Q1 | Q2 | Q3 | Q4 | |
|---|---|---|---|---|---|---|---|---|---|---|
| 0.1 | 96.0 | 88.0 | 84.0 | 60.0 | 82.0 | 3.44 | 17.60 | 31.20 | 43.68 | 23.98 |
| 0.3 | 96.0 | 92.0 | 80.0 | 68.0 | 84.0 | 7.80 | 13.36 | 31.08 | 36.56 | 22.20 |
| 0.5 | 100.0 | 92.0 | 84.0 | 64.0 | 85.0 | 7.52 | 16.44 | 30.48 | 38.92 | 23.34 |
| 0.7 | 100.0 | 88.0 | 80.0 | 76.0 | 86.0 | 5.76 | 13.92 | 28.68 | 32.92 | 20.32 |
| 0.9 | 100.0 | 88.0 | 76.0 | 76.0 | 85.0 | 9.32 | 9.92 | 32.16 | 27.72 | 19.78 |
| 2.0 | 96.0 | 84.0 | 72.0 | 76.0 | 82.0 | 5.36 | 16.32 | 31.96 | 36.60 | 22.56 |

### C.6.2 SEARCH STRATEGY PARAMETERS

Table 22 evaluates the UCB exploration parameter $c$, which controls the exploration-exploitation trade-off in tree search. The optimal value $c = 0.5$ maintains consistent performance across all complexity levels. Higher values ($c \geq 1.0$) cause excessive exploration, particularly harming high-complexity molecules (Q4: drops to 52% at $c = 5.0$).

Table 22: UCB exploration parameter $c$ impact.

| $c$ Value | Solve Rate (%) | | | | Avg SR | Iterations | | | | Avg Iter. |
|---|---|---|---|---|---|---|---|---|---|---|
| | Q1 | Q2 | Q3 | Q4 | | Q1 | Q2 | Q3 | Q4 | |
| 0.2 | 96.0 | 88.0 | 88.0 | 72.0 | 86.0 | 5.36 | 13.88 | 27.72 | 35.72 | 20.67 |
| 0.5 | 100.0 | 88.0 | 80.0 | 76.0 | 86.0 | 5.76 | 13.92 | 28.68 | 32.92 | 20.32 |
| 1.0 | 100.0 | 88.0 | 76.0 | 72.0 | 84.0 | 3.60 | 22.28 | 32.64 | 28.60 | 21.78 |
| 1.414 | 100.0 | 88.0 | 72.0 | 60.0 | 80.0 | 6.28 | 15.88 | 35.36 | 42.48 | 25.00 |
| 2.0 | 92.0 | 92.0 | 76.0 | 68.0 | 82.0 | 9.24 | 16.72 | 31.48 | 39.40 | 24.21 |
| 5.0 | 96.0 | 84.0 | 72.0 | 52.0 | 76.0 | 5.84 | 18.76 | 50.52 | 37.60 | 28.18 |

### C.6.3 REWARD FUNCTION WEIGHTS

Table 23 analyzes the availability weight $\alpha$ in the reward function. The optimal value $\alpha = 0.4$ balances immediate building block availability with long-term synthesis feasibility. Pure feasibility scoring ($\alpha = 0.0$) degrades performance by 3%, while pure availability scoring ($\alpha = 1.0$) shows 7% reduction, confirming that both components are essential for effective search guidance.

Table 23: Availability weight $\alpha$ impact on performance metrics.

| $\alpha$ value | Solve Rate (%) | | | | Avg SR | Iterations | | | | Avg Iter. |
|---|---|---|---|---|---|---|---|---|---|---|
| | Q1 | Q2 | Q3 | Q4 | | Q1 | Q2 | Q3 | Q4 | |
| 0.0 | 100.0 | 92.0 | 72.0 | 68.0 | 83.0 | 4.96 | 13.56 | 23.16 | 30.40 | 18.02 |
| 0.2 | 100.0 | 88.0 | 76.0 | 72.0 | 84.0 | 4.72 | 9.56 | 32.04 | 43.00 | 22.33 |
| 0.4 | 100.0 | 88.0 | 80.0 | 76.0 | 86.0 | 5.76 | 13.92 | 28.68 | 32.92 | 20.32 |
| 0.6 | 96.0 | 92.0 | 80.0 | 68.0 | 84.0 | 6.20 | 14.81 | 30.23 | 34.69 | 21.47 |
| 0.8 | 96.0 | 92.0 | 76.0 | 68.0 | 83.0 | 8.64 | 10.32 | 29.36 | 35.00 | 20.83 |
| 1.0 | 92.0 | 88.0 | 72.0 | 64.0 | 79.0 | 7.80 | 15.25 | 32.44 | 38.25 | 23.40 |

### C.7 ROUTE CHARACTERISTICS ANALYSIS

### C.7.1 ROUTE LENGTH DISTRIBUTION

Tables 24 and 25 show how route length correlates with molecular complexity across all benchmarks. Complex molecules require longer routes—average length increases from 3.64 steps (Q1) to 6.86 steps (Q4) on Pistachio Hard. Notably, 76% of simple molecules (Q1) are solved in 1-4 steps, while complex molecules (Q4) predominantly require 5-8 steps. USPTO-190 shows similar patterns but with consistently longer routes (5.52-6.35 steps), reflecting its focus on multi-step pharmaceuticals rather than simpler organic molecules.

### C.8 SUCCESS CASE: COMPLEX NATURAL PRODUCT

We provide AOT* visualizations of successful cases across diverse pharmaceutical-relevant molecules with high synthetic complexity. Figures 13-15 showcase the effectiveness on drug-like molecules from the USPTO-190 dataset, containing diverse functional groups such as nitriles, oxiranes, indoles, and iodinated aromatics. These pharmaceutically relevant structures represent significant synthetic challenges, yet AOT* consistently identifies multiple viable routes through focused tree expansion. The compact tree structures, characterized by strategic branching patterns and high-confidence pathways, demonstrate the efficiency gains from LLM-guided generation. AOT*'s ability

Table 24: Route length distribution by molecular complexity for challenging benchmarks.

| Metric | USPTO-190 | | | | Pistachio Hard | | | |
|---|---|---|---|---|---|---|---|---|
| | Q1 | Q2 | Q3 | Q4 | Q1 | Q2 | Q3 | Q4 |
| Solve Rate (%) | 100.0 | 85.1 | 81.2 | 78.7 | 100.0 | 88.0 | 80.0 | 76.0 |
| Avg. Length | 5.52 | 5.71 | 6.33 | 6.35 | 3.64 | 4.88 | 5.41 | 6.86 |
| 1-4 steps (%) | 47.9 | 29.3 | 32.5 | 32.5 | 76.0 | 45.8 | 13.6 | 36.4 |
| 5-8 steps (%) | 37.5 | 56.1 | 47.5 | 42.5 | 24.0 | 54.2 | 59.1 | 59.1 |
| 9+ steps (%) | 14.6 | 14.6 | 20.0 | 25.0 | 0.0 | 0.0 | 27.3 | 4.5 |

Table 25: Route length distribution by molecular complexity for simpler benchmarks.

| Metric | USPTO Easy | | | | Pistachio Reachable | | | |
|---|---|---|---|---|---|---|---|---|
| | Q1 | Q2 | Q3 | Q4 | Q1 | Q2 | Q3 | Q4 |
| Solve Rate (%) | 100.0 | 100.0 | 100.0 | 100.0 | 100.0 | 100.0 | 97.3 | 97.4 |
| Avg. Length | 2.14 | 3.29 | 3.56 | 4.53 | 3.45 | 3.97 | 4.30 | 4.46 |
| 1-4 steps (%) | 90.0 | 79.2 | 79.6 | 53.2 | 76.3 | 64.9 | 72.2 | 62.2 |
| 5-8 steps (%) | 10.0 | 12.5 | 16.3 | 36.2 | 21.1 | 29.7 | 27.8 | 32.4 |
| 9+ steps (%) | 0.0 | 8.3 | 4.1 | 10.6 | 2.6 | 5.4 | 0.0 | 5.4 |

to balance exploration and exploitation is particularly evident in how it handles structural complexity—maintaining synthetic feasibility while discovering creative disconnection strategies through the integration of generative models with systematic tree search.

The Pistachio Hard dataset examples (Figures 16-18) further validate AOT*'s ability to handle challenging targets including molecules featuring complex heterocyclic scaffolds, multiple stereocenters, and elaborate ring systems. The search trees reveal how AOT* efficiently navigates vast chemical spaces through strategic pathway generation rather than exhaustive enumeration. Notably, AOT* successfully decomposes these intricate structures—ranging from triazole-piperidine conjugates to spirocyclic fluorinated fragments—into commercially available building blocks while maintaining reasonable synthesis depths. The visualizations illustrate the framework's adaptive search behavior, where computational resources are allocated based on molecular complexity, enabling both rapid convergence for simpler substructures and thorough exploration for challenging disconnections.

## C.9 Failure Analysis

While AOT* demonstrates strong performance overall, certain molecules with exceptionally high synthetic complexity expose current limitations. Figures 19-21 illustrate challenging cases where extensive exploration fails to complete synthesis routes from Pistachio Hard and USPTO-190. All three failures exhibit similar patterns: dense and deep search trees with extensive branching, and numerous reaction attempt. All explore many pathways but struggles to find routes to available building blocks, suggesting insufficient guidance for prioritizing promising directions.

These failures highlight clear improvement opportunities: incorporating domain-specific reaction knowledge, developing escape mechanisms from unproductive search regions, and enhancing strategic flexibility when standard approaches fail. However, such limitations affect only a small fraction of targets. AOT* successfully solves the vast majority of complex pharmaceutical molecules, demonstrating robust performance across diverse structural classes. By combining LLM-guided generation with systematic tree search, the framework achieves both efficiency and reliability—offering chemists a powerful tool that discovers novel synthetic strategies while maintaining chemical validity. The algorithm's ability to handle molecules ranging from simple heterocycles to elaborate natural products validates its practical utility for automated synthesis planning.

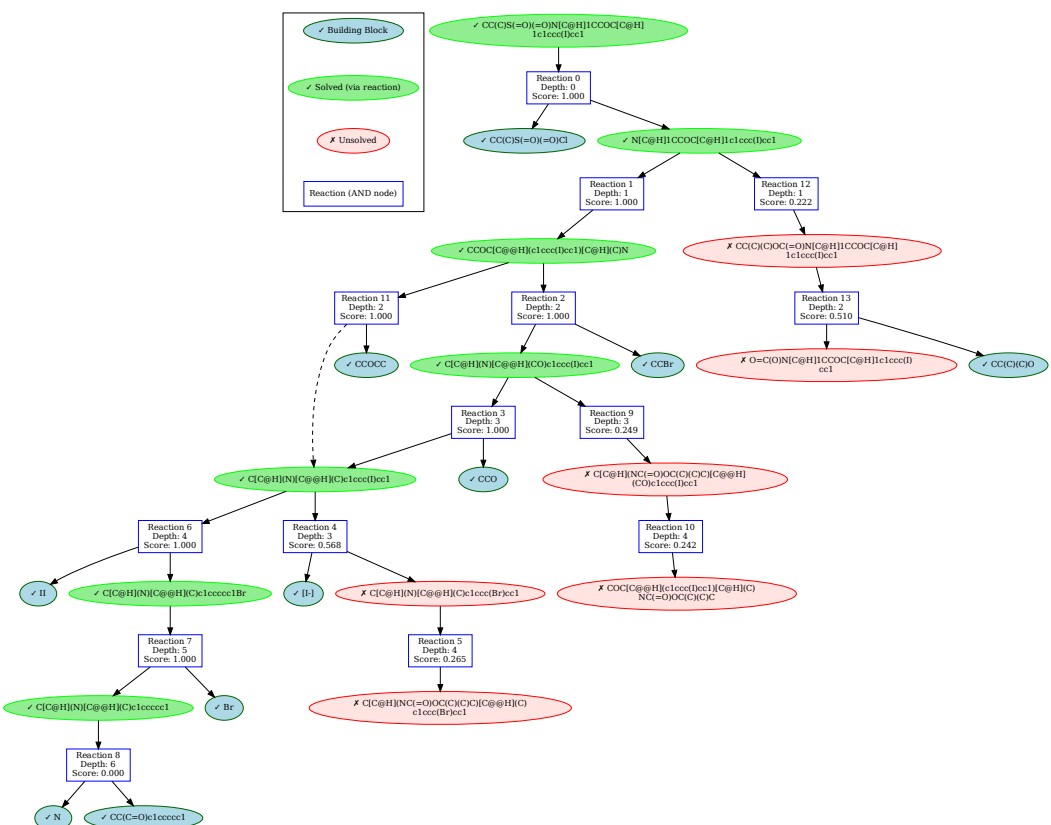

Figure 13: CC(C)S(=O)(=O)N[C@H]1CCOC[C@H]1c1ccc(I)cc1, USPTO-190, Visualization of AOT* search tree.

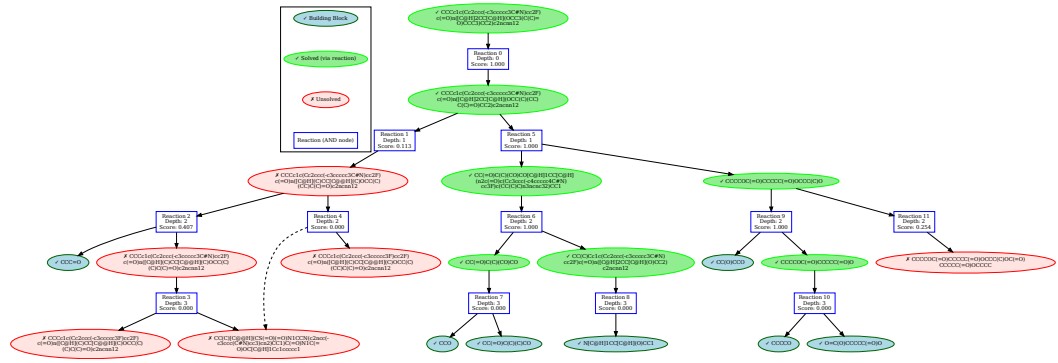

Figure 14: CCCc1c(Cc2ccc(-c3ccccc3C#N)cc2F)c(=O)n([C@H]2CC[C@H](OCC3(C(C)=O)CCC3)CC2)c2ncnn12, USPTO-190, Visualization of AOT* search tree.

# D  LIMITATIONS AND FUTURE WORK

Despite AOT*'s efficiency improvements, several limitations remain. The framework depends on the underlying LLM's chemical knowledge, which may not capture specialized transformations well. Complex natural products can still cause unproductive search expansions, indicating that tree search cannot fully compensate for gaps in chemical understanding. Moreover, the current framework lacks mechanisms for controllable multi-objective search and uncertainty quantification—features essential for deployment where failed reactions incur significant costs. Future work should address these limitations by developing approaches that generalize beyond training distributions, incorporate

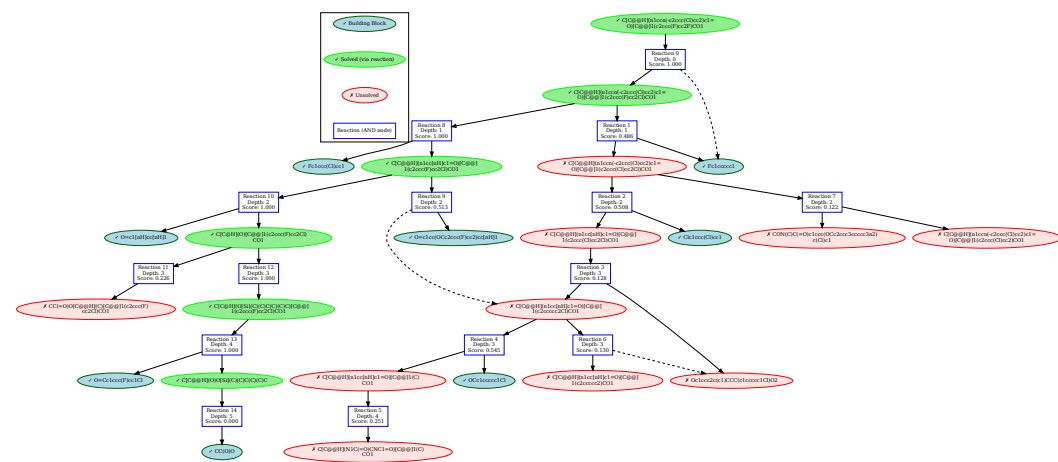

Figure 15: C[C@@H](n1ccn(-c2ccc(Cl)cc2)c1=O)[C@@]1(c2ccc(F)cc2F)CO1, USPTO-190, Visualization of AOT* search tree.

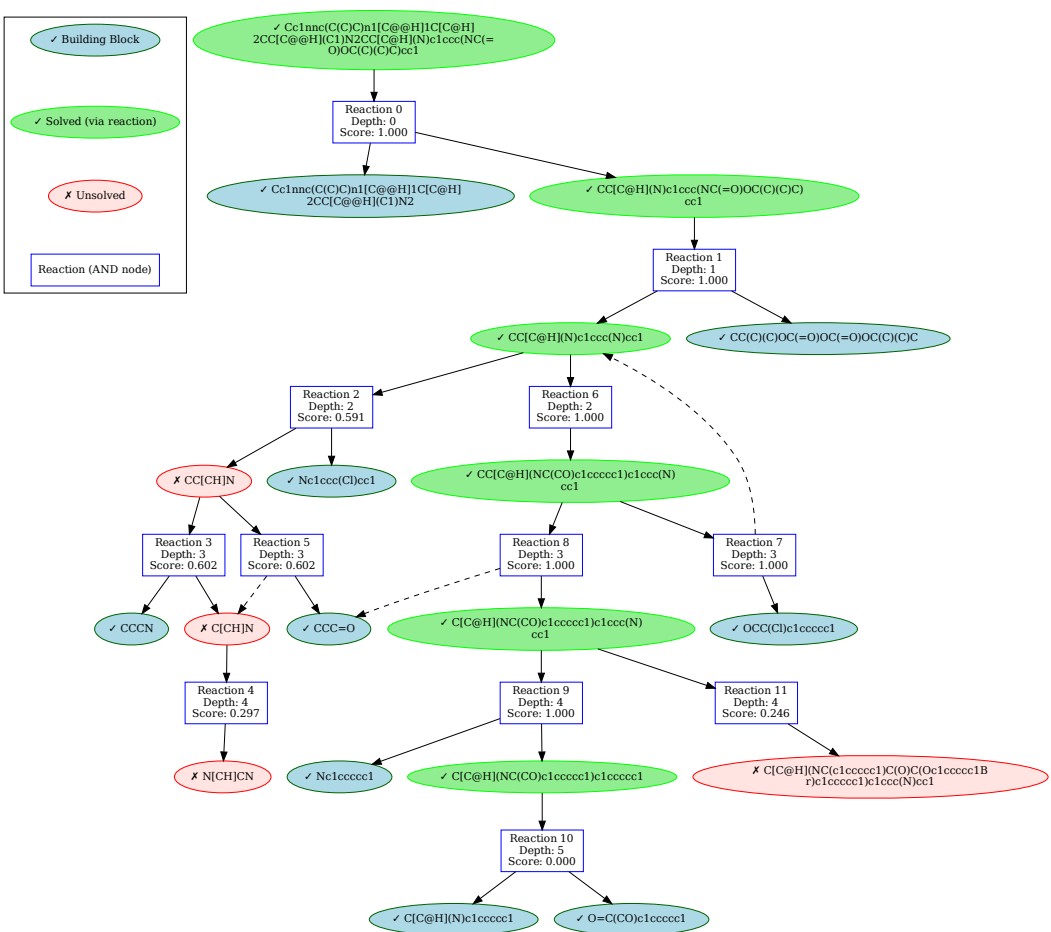

Figure 16: Cc1nnc(C(C)C)n1[C@@H]1C[C@H]2CC[C@@H](C1)N2CC[C@H](N)c1ccc(NC(=O)OC(C)(C)C)cc1, Pistachio Hard, Visualization of AOT* search tree.

controllable generation for diverse synthetic priorities, and integrate uncertainty estimates to guide practical decision-making in chemical synthesis.

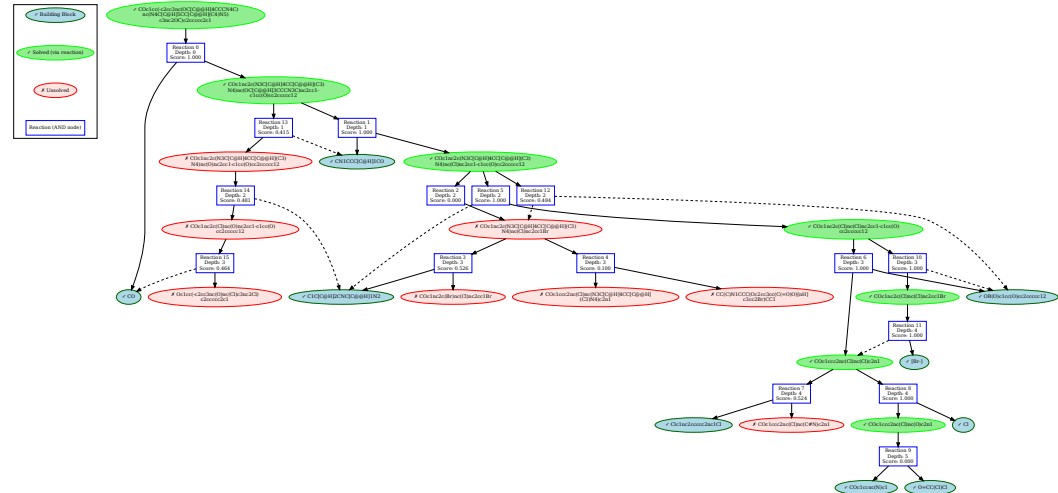

Figure 17: COc1cc(-c2cc3nc(OC[C@@H]4CCCN4C)nc(N4C[C@H]5CC[C@@H](C4)N5)c3nc2 OC)c2ccccc2c1, Pistachio Hard, Visualization of AOT* search tree.

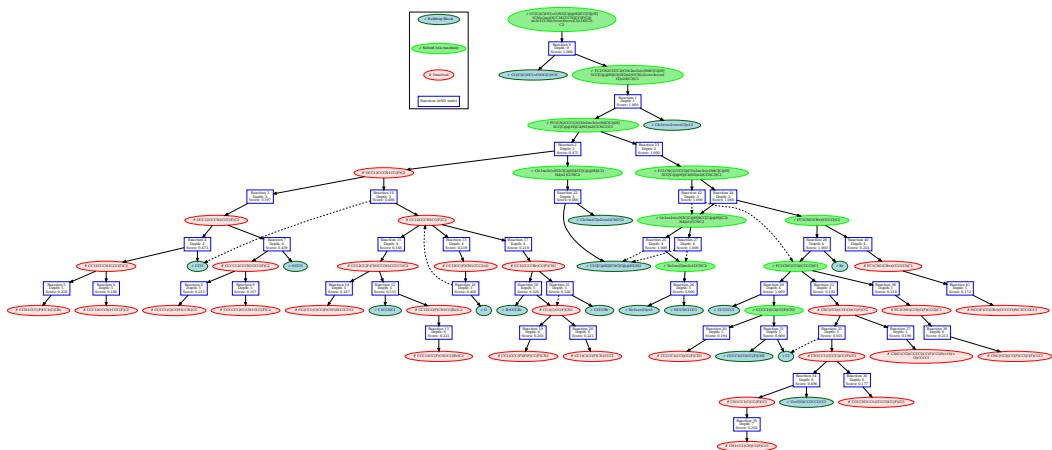

Figure 18: CC(C)(C)OC(=O)N1[C@@H]2CC[C@H]1CN(c1nc(OCC34CCCN3CC(F)C4)nc3c1C CN(c1cccc4cccc(Cl)c14)C3)C2, Pistachio Hard, Visualization of AOT* search tree.

Future work could address these limitations through several directions. Development of specialized chemical LLMs through distillation from general models could significantly reduce computational costs while maintaining performance—our experiments show that general-purpose LLMs incur substantial token overhead that specialized models might avoid. Enhanced reasoning capabilities integrated with tree search could help the system recognize and articulate when it ventures into uncertain chemical territory, potentially reducing unproductive expansions. Adaptive search strategies that dynamically adjust between exploration and exploitation based on molecular complexity could better allocate computational resources. Finally, incorporating multi-objective optimization into the tree search framework would enable practitioners to specify trade-offs between synthesis length, yield, and safety constraints, making the system more applicable to real-world synthesis planning where such considerations are paramount.

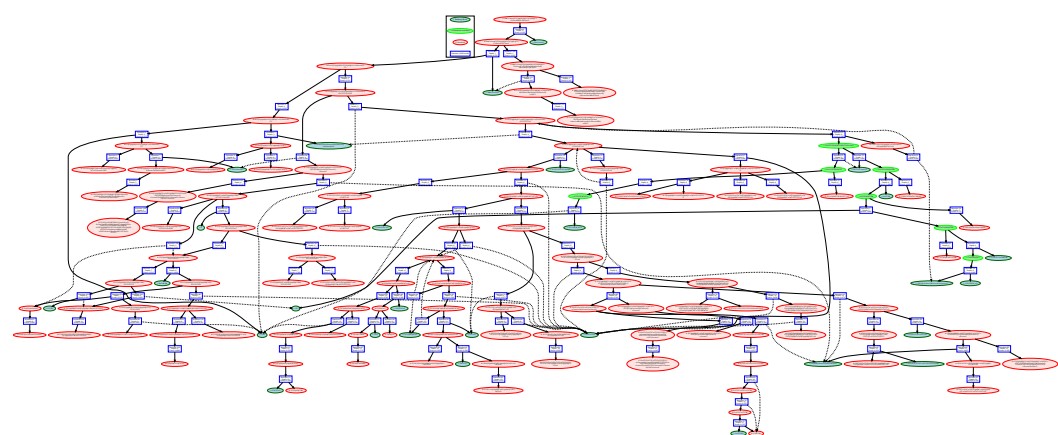

Figure 19: Failure case: COCCCc1cc(CN(C(=O)[C@H]2CN(C(=O)OC(C)(C)C)CC[C@@H]2c2ccc(OCCOc3c(Cl)cc(C)cc3Cl)cc2)C2CC2)cc(OCCOC)c1.

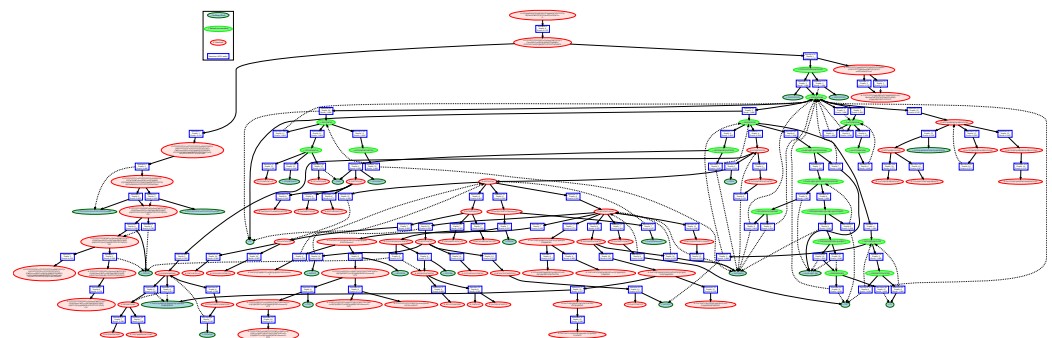

Figure 20: Failure case: C[C@@H](O)C[C@H]1OC[C@@H](C2CCCCC2)N(c2cc(C#CC(C)(C)C)sc2C(=O)O)C1=O.

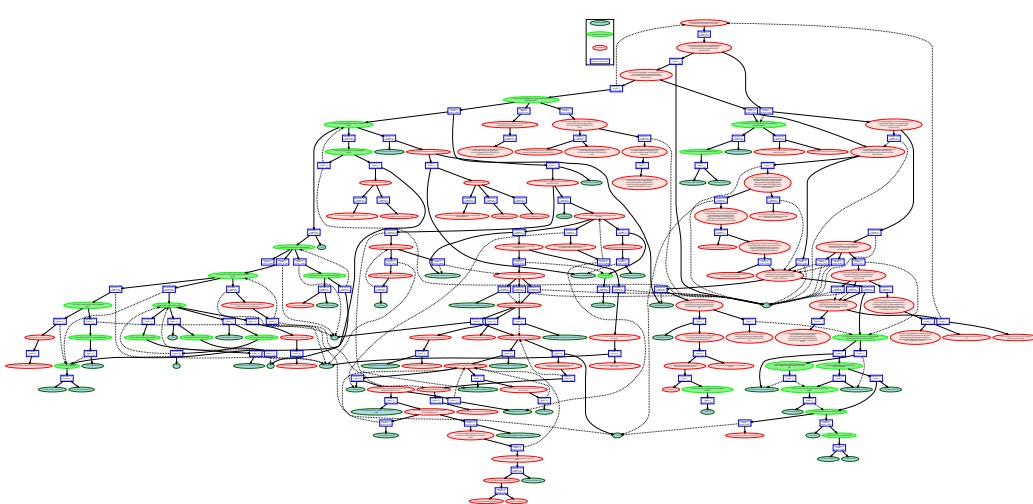

Figure 21: Failure case: C[Si](C)(C)CCOCn1cc(C2CCc3c(C(=O)O)nn(COCC[Si](C)(C)C)c3C2)cn1.

