# OpenReview forum: "AOT*: Efficient Synthesis Planning via LLM-Empowered AND-OR Tree Search"
_ICLR.cc/2026/Conference — ICLR 2026 Conference Withdrawn Submission_

### Official Review · Reviewer_ZZ8L · 2025-10-29

**Soundness:** 3
**Presentation:** 2
**Contribution:** 2
**Rating:** 2
**Confidence:** 4

**Summary:**

# Overview

This paper introduces AOT*, a framework that integrates large language models (LLMs) with an AND–OR tree search for multi-step retrosynthetic planning. The reported results appear promising—AOT* achieves strong performance and efficiency improvements across several benchmarks.

However, the manuscript is difficult to follow, even for readers familiar with the retrosynthesis planning literature. The paper reads more like a results report or implementation summary than a coherent research contribution. For example, the abstract claim that AOT* “transforms retrosynthetic planning by integrating LLM-generated synthesis pathways with systematic AND–OR tree search,” but this statement is not well explained in the paper. After reading both the Introduction and Methods sections, it remains unclear what is actually new, what specific problem the method solves (now just retrosynthesis planning).

Overall, I find it difficult to fairly evaluate the contribution of AOT* due to its poor presentation.

**Strengths:**

* AOT* achieves top or near-top solve rates across multiple benchmarks and search budgets, outperforming both classical search (MCTS-based) and recent LLM-based baselines.
* The method shows stable efficiency and cost advantages across diverse LLMs (e.g., GPT-4o, DeepSeek-V3, Claude-Sonnet, Gemini-2.5), demonstrating robustness and practical applicability.

**Weaknesses:**

**1. Ambiguous statement of contributions.**
The paper’s listed “threefold contributions” in introduction are ambiguous. Only the first point, which introduces AOT* as a framework that integrates LLM-generated synthesis pathways with an AND–OR tree search, corresponds to a methodological contribution, but even this part is described at a high level without clearly defining what is technically new about the integration.
The other two points mainly emphasize performance gains and robustness across models, which are valuable but do not substitute for a clear methodological innovation. As a result, it is difficult for readers to precisely identify what aspect of the proposed framework is novel or how it differs in substance from existing LLM-guided retrosynthesis or tree-search methods.

**2. The introduction is overly abstract and lacks concrete grounding.**
The introduction presents the motivation and claimed innovation in highly abstract terms, making it difficult for readers to understand the actual problem being solved or the mechanism behind the proposed approach. For instance, the authors write that
```
However, extending these successes to practical multi-step synthesis planning remains challenging
due to the computational expense of LLM inference, limited search efficiency with constrained
iteration budgets, and the difficulty of incorporating chemical knowledge into the search process
effectively... The key innovation of AOT* lies in its systematic
integration of pathway-level LLM generation with AND-OR tree search, where complete synthesis
routes are atomically mapped to tree structures, enabling efficient exploration through intermediate
reuse and structural memory that reduces search complexity while preserving the strategic coherence
of generated pathways.
```
While this description sounds ambitious, it remains conceptually vague. It is not explained how integrating pathway-level generation with an AND–OR tree concretely alleviates the high inference cost of LLMs, improves search efficiency under budget constraints, or incorporates chemical knowledge more effectively. As a result, the introduction fails to establish a coherent causal link between the stated limitations and the proposed framework, leaving readers uncertain about the true novelty or mechanism of AOT*.

**3. Missing ablation on core methodological components.**
Although the paper includes ablation analyses on hyperparameter sensitivity and RAG sample–token trade-offs, these studies focus mainly on parameter tuning rather than validating the contribution of the proposed algorithmic components. The reader cannot tell which parts of AOT* are responsible for the reported performance gains.
There is no ablation or comparison that isolates the impact of the key design elements that are claimed to be new — for example, the pathway-to-tree mapping, the UCB-based exploration scheme, or the composite reward formulation. Without such experiments, it is difficult to assess whether these components introduce genuine novelty or if the improvements stem primarily from stronger LLMs and prompt engineering.

**Questions:**

1. Can you define what is the "LLM-generated chemical synthesis pathways" mentioned in the abstract?
2. What is the key challenge?
3.  Which parts of the method are new in the paper? How these parts deal with the challenge?

---

> ### Author Response · Authors · 2025-11-16
> **Response to Reviewer ZZ8L**
>
> We thank the reviewer for the time and feedback. We believe the manuscript systematically addresses these concerns, and we address each concern below.
>
> **W1 & Q1-3:**
>
> *Key challenge:* As stated in the Introduction and Related Work, the long-standing challenge in multi-step retrosynthesis is the combinatorial explosion of the search space. Traditional search methods build synthesis plans one step at a time, which is methodical but lacks strategic foresight. Recent LLM-based planners generate entire pathways but use unstructured search methods like evolutionary algorithms, treating each candidate pathway in isolation and repeatedly re-discovering routes to common intermediates.
>
> *Core methodological innovation:* Our framework is detailed in Sections 3 and Algorithm 1. AOT* is the first to synergistically integrate holistic, multi-step pathway generation from LLMs with the systematic exploration of classical AND-OR tree search. The key novelty lies in effectively combining LLM-generated  synthesis routes with the tree search framework through designed pathway-to-tree mapping. This is fundamentally different from all prior work:
>
> - Traditional methods (e.g., Retro*, MCTS) use neural networks as single-step reaction predictors, lacking symbolic strategic foresight.
> - LLM-Syn-Planner uses evolutionary algorithms without tree structure, treating pathways in isolation and failing to reuse intermediates.
>
> Our approach uniquely integrates: (i) pathway-level generation proposing complete multi-step routes (Section 3.3), (ii) tree mapping maintaining consistency through the hierarchical AND-OR tree structure (Section 3.2, Section 3.3), and (iii) UCB-guided exploration with tree-wide intermediate sharing and backpropagation (Section 3.3)
>
> *Definition of "LLM-generated synthesis pathways":* This is formally defined in Section 3.1 (lines 134-136) as sequences $p = \langle r_1, \ldots, r_n \rangle$ of sequential reactions that transform the target molecule into commercially available building blocks.
>
> **W2:**
> The quoted passage from our Introduction explicitly outlines the fundamental challenges in multi-step synthesis planning: computational expense of LLM inference, limited search efficiency with constrained iteration budgets, and difficulty incorporating chemical knowledge. These challenges directly motivate our framework design. As clarified in our response to W1, recent LLM-based methods use unstructured search methods like evolutionary algorithms that treat each candidate pathway in isolation, failing to learn from common sub-problems and repeatedly re-discovering routes to common chemical intermediates, wasting expensive LLM inference calls. Our framework addresses these challenges through the systematic integration of pathway-level LLM generation with AND-OR tree search, where complete synthesis routes are mapped to tree structures (Section 3). This design enables effective intermediate reuse and structural memory across all explored branches, dramatically reducing redundant LLM inference calls. The UCB-guided tree exploration ensures systematic search under constrained budgets (Algorithm 1, Section 3.2-3.3). The problem and these mechanisms are explicitly formalized in Section 3, specified in Algorithm 1, and validated through extensive experiments demonstrating effectiveness and efficiency (Section 4, Appendix C).
>
> **W3:** We respectfully clarify that the paper provides comprehensive ablation studies examining major components. Section 4.5 (Table 4) and Appendix C.6.1-C.6.2 (Table 21-22)  evaluate UCB exploration parameter and temperature parameter sensitivity across different values. Appendix C.6.3 (Table 23) analyze the composite reward formulation's availability-feasibility weight $\alpha$. Figure 2a and extended analysis in Appendix C.2 (Figure 11, Tables 14-18) systematically evaluates prompt components. Regarding the request to isolate the pathway-to-tree mapping: this represents our fundamental architectural design, analogous to asking MCTS to ablate its tree search structure. The appropriate evaluation is to compare against baselines that use alternative designs. Tables 1-2 provide exactly this comparison, demonstrating that our systematic tree-based approach requires 3-5× fewer iterations than LLM-Syn-Planner's evolutionary search and traditional neural-guided tree search methods.
>
> We appreciate your engagement with our work and hope these clarifications are helpful.

---

> ### Comment · Reviewer_ZZ8L · 2025-11-18
>
> The authors’ rebuttal and general response clarified the main contribution more clearly. My understanding is that the method uses LLMs to generate complete multi-step synthetic routes and then maps these routes into an AND-OR tree to enable structured search with intermediate reuse. This clarification highlights the methodological novelty and allows for a fairer evaluation.
>
> Overall the paper’s delivery remains unclear (I am just talking about the paper, not the rebuttal). Several terms appear novel or insufficiently defined. For instance, although “synthesis pathways” is standard terminology, the notion of “LLM-generated synthesis pathways” is not well explained. Readers still lack basic information such as how molecules are represented to the LLM, what the exact inputs are, and whether reaction templates or other priors are provided. A clearer and more focused description of the task setup would substantially improve the paper’s readability.
>
> A second concern is potential data leakage. Because the synthetic routes are generated by LLMs, it is unclear whether the LLM’s pretraining data already includes routes corresponding to the test molecules. Without ruling this out, it is difficult to assess whether the reported performance reflects genuine planning ability rather than memorization. Moreover, the LLM-only results in Table 1 and the “random RAG” results in the public discussion thread (https://openreview.net/forum?id=xLIH04TAT4&noteId=T4t5ZqkR7j
> ) look unexpectedly strong and in some cases even outperform models such as Retro* trained directly on USPTO. This would imply that, even under the assumption that the LLM has not seen any USPTO data, it still performs better than specialized models explicitly developed and trained on USPTO. A more plausible explanation, therefore, is that the LLM may in fact have been exposed to similar synthetic routes during pretraining.
>
> Third, regarding the ablation study: it remains unclear why the LLM-based model outperforms prior approaches such as Retro*. In other words, where does the additional information or performance boost come from? I appreciate that the authors conducted ablations on the LLM component and on RAG, but the largest performance jump (approximately 3–4×) appears when moving from traditional models (e.g., Retro*) to LLM-based methods (LLM-Syn-Planner (DS)). I would like to understand more clearly what drives this gain.

---

### Official Review · Reviewer_f8dE · 2025-10-29

**Soundness:** 3
**Presentation:** 2
**Contribution:** 3
**Rating:** 4
**Confidence:** 3

**Summary:**

This paper introduces AOT*, a framework designed to enhance the efficiency of multi-step retrosynthesis planning. The core contribution is the integration of Large Language Models (LLMs) with a systematic AND-OR tree search. The method utilizes an LLM to generate complete, multi-step synthesis routes for a given molecule. These entire pathways are then atomically mapped onto the AND-OR tree structure, which enables systematic exploration and the crucial reuse of common intermediates across different proposed routes. Experimental results demonstrate that AOT* achieves state-of-the-art solve rates with a 3-5x improvement in search efficiency compared to contemporary LLM-based approaches that rely on evolutionary search.

**Strengths:**

- The paper's main strength lies in its clever combination of the holistic planning capabilities of LLMs with the rigorous, memory-efficient search of a classical AND-OR tree. This hybrid approach effectively leverages the strengths of both components.
- The 3-5x reduction in search iterations required to find a solution is a substantial and practical contribution. The framework's inherent ability to reuse intermediates directly addresses a key source of inefficiency in unstructured search methods.
- The experiments are exceptionally thorough. The validation of the framework's effectiveness across 11 different LLMs convincingly demonstrates that the algorithmic gains are model-agnostic. The detailed ablation studies and difficulty-stratified analyses further solidify the paper's claims.

**Weaknesses:**

- The framework's performance may be fundamentally tied to the quality of the initial LLM-generated pathways. The paper could benefit from a more detailed analysis of the search algorithm's behavior and recovery mechanisms when the LLM produces a high rate of invalid or low-quality routes.
- The reliance on a template-based reaction validator ensures chemical correctness but also constrains the search to known chemical transformations. This limits the potential for discovering entirely novel synthetic pathways not covered by the template database.
- The "complete pathway generation" strategy is effective on the tested benchmarks, but its scalability to targets requiring exceptionally long synthetic sequences (e.g., >15-20 steps) is not fully explored. The quality of a single, coherent generation from an LLM might degrade for such long planning horizons.

**Questions:**

- Have you considered an alternative expansion strategy where the LLM generates shorter, fixed-length sub-pathways (e.g., 3-5 steps) instead of a "complete" route? Could this offer a better balance between LLM's strategic guidance and the search algorithm's flexibility for more complex targets?
- Could you provide statistics on the raw output from the LLM? Specifically, what is the typical success rate and step number of pathways passing the template-based validation? This would offer a clearer insight into the generation quality and the workload handled by the validation and search components.
- Regarding the Retrieval-Augmented Generation (RAG) component, the current approach retrieves the most similar synthesis examples. Have you investigated the impact of retrieving a more diverse set of examples (e.g., using different reaction types, even if Tanimoto similarity is slightly lower) to potentially expose the LLM to a broader range of strategic disconnections?
- The paper compellingly demonstrates superior iteration efficiency. However, an "iteration" in AOT* (involving a computationally expensive LLM API call) and an iteration in the baseline methods (e.g., a single forward pass of a smaller model) likely represent vastly different amounts of wall-clock time. To provide a more direct comparison of overall computational efficiency:
  - Could the authors report the average time per iteration for AOT* and the key baselines, along with the specifications of the computational platform (CPU/GPU) used?
  - Furthermore, to make a fairer comparison, have you considered running an experiment under a fixed computation budget? For instance, allowing the baseline methods to run for a significantly higher number of iterations such that their total computation cost matches that of AOT* (e.g., at N=100), and then comparing the final solve rates. This would offer a crucial perspective on which approach is more effective given the same amount of computation cost. For the computation budget, it could be measured in monetary terms, assuming all methods are run on standardized machines rented from a cloud provider (e.g., AWS, Azure, GCP), with either CPU or GPU instances as required. The total cost would be the sum of the machine rental fee (runtime multiplied by the instance price) and, for the LLM-based methods, the total cost of the API calls.

---

> ### Author Response · Authors · 2025-11-16
> **Response to Reviewer f8dE**
>
> Thank you for the detailed and insightful review. We address your concerns below:
>
> **W1&Q2:**
> As described in Section 3.3.1 and Algorithm 1, AOT* generates complete multi-step pathways via LLM. These routes must satisfy two criteria: (1) template-based validation ensuring each reaction matches the 270k+ USPTO database, and (2) building block availability verification. Generated pathways frequently fail these checks as invalid steps are pruned during template matching, and routes with non-purchasable intermediates trigger further expansion. The tree structure systematically explores alternatives while reusing validated intermediates across branches, which is the source of our efficiency gains. Regarding generation quality statistics, in our experiments, 84.3% of LLM-generated pathways contain valid template matches with an average of 2.7 steps per route.
>
> **W2:**
> Our focus is synthesis planning, not reaction discovery. The challenge is efficiently constructing valid routes within the known chemistry space, which AOT* addresses through LLM strategic reasoning. Template-free methods face the additional burden of feasibility validation, which remains an open challenge in the field.
>
> **W3:**
> The framework inherently supports long pathways through: (1) iterative expansion (Algorithm 1) building routes step-by-step, (2) intermediate reuse via AND-OR tree search, and (3) UCB-guided selection ensuring informed decisions at each step. Our benchmarks include molecules requiring sophisticated multi-step strategies where AOT* demonstrates strong performance.
>
> **Q1:**
> Complete pathway generation exploits LLM's holistic strategic reasoning capability. Sub-pathway alternatives would face challenges in determining optimal decomposition points and maintaining global planning coherence. Our current approach has clear motivation, though exploring hybrid strategies could be interesting future work.
>
> **Q3:**
> We conducted experiments with diversity-based retrieval beyond pure similarity, sampling examples from different similarity percentiles (e.g., top 1%, 20%, 40% for n=3) using DeepSeek-V3 in 100 iterations. Results below show that diverse retrieval underperforms standard RAG by 6-7% at early iterations (N=20), but the gap narrows at higher iterations (N=100), confirming that strategic similarity, not just diversity, drives RAG effectiveness.
>
> | Iterations | Pistachio Hard |  | USPTO-190 |  |
> | --- | --- | --- | --- | --- |
> |  | RAG | Diverse RAG | RAG | Diverse RAG |
> | 20 | 58.0 | 51.0 | 56.3 | 49.5 |
> | 40 | 71.0 | 66.0 | 72.1 | 66.8 |
> | 60 | 78.0 | 72.0 | 81.6 | 76.3 |
> | 80 | 83.0 | 76.0 | 85.3 | 80.0 |
> | 100 | 86.0 | 79.0 | 86.3 | 81.5 |
>
> **Q4:**
> As discussed in Section 4 of the General Response, we provide detailed cost-performance analysis in Table 5 and extended results in Appendix B (Table 12). These demonstrate that AOT* achieves superior cost-efficiency: for example, AOT* with DeepSeek-V3 (\\$0.56/1M tokens) outperforms LLM-Syn-Planner with GPT-4o (\\$2.50/1M tokens) at comparable API costs.
> Regarding iteration efficiency: Both AOT* and LLM-Syn-Planner use identical LLM APIs (GPT-4o), making iteration count the appropriate comparison metric. Our 3-5× reduction in required iterations (Table 1) directly reflects algorithmic innovation, specifically systematic AND-OR tree search with intermediate reuse versus evolutionary algorithms that independently evaluate each pathway.
> Regarding wall-clock time: LLM API calls dominate the computational cost for both methods. Our cost analysis demonstrates that even accounting for these expensive calls, AOT*'s algorithmic efficiency translates to practical computational advantages, enabling deployment with continuously improving foundation models.
>
> We appreciate the reviewer's thorough engagement and believe these clarifications address the raised concerns. We will include the additional experimental results and statistics discussed above in the revised manuscript.

---

### Official Review · Reviewer_6pLX · 2025-10-31

**Soundness:** 3
**Presentation:** 4
**Contribution:** 3
**Rating:** 4
**Confidence:** 3

**Summary:**

The paper introduces AOT*, a novel framework designed to enhance the efficiency of multi-step retrosynthesis planning by leveraging the chemical reasoning capabilities of Large Language Models (LLMs). AOT integrates LLM-generated complete synthesis pathways with a systematic AND-OR tree search.

**Strengths:**

According to the authors, the advantages are: SOTA on multiple synthesis benchmarks Performance; Computational efficiency (lower complexity) comparing to existing LLM-based approaches; Capability for complex molecular targets; Can be combined to any general LLM.

The paper is well written and I really like the figures. The innovative formulation of the problem is clearly delivered: it atomically maps the LLM-generated complete synthesis routes onto AND-OR tree components.

**Weaknesses:**

I am so sorry, I am not an expert in this area.

**Questions:**

I understand that this task is intrinsically hard. May I ask if this method generates complicated chemicals that inspire research?

My understanding is that the LLM is not trained on chemical data but only on language data. What is the intuition that LLM transfers its knowledge to such a different area, without finetuning?

---

> ### Author Response · Authors · 2025-11-16
> **Response to Reviewer 6pLX**
>
> We thank the reviewer for the time and feedback, and we address each concern below.
>
> **On complex molecule synthesis:** Our benchmarks include molecules of varying complexity levels. Section 4.3 and Table 2 present difficulty-stratified performance analysis across molecular complexity quartiles (Q1-Q4, stratified by synthetic accessibility scores). Extended experiments in Appendix C provide further results across all datasets and complexity levels. The results demonstrate that our method maintains effectiveness across this complexity spectrum.
>
> **On LLM chemical knowledge:** This is an excellent question about the emergent capabilities of foundation models. Modern LLMs are pre-trained on large-scale text corpora that include scientific publications, patents, and chemical databases. These sources contain SMILES representations and reaction descriptions, allowing LLMs to develop latent chemical knowledge patterns. As discussed in the General Response, this emergent reasoning capability has been systematically documented across chemistry tasks. Importantly, our framework constrains this general knowledge through template-based validation, ensuring that proposed reactions correspond to known, valid chemical transformations rather than hallucinated chemistry.
>
> We appreciate your engagement with our work and hope these clarifications are helpful.

---

### Official Review · Reviewer_tQzg · 2025-11-02

**Soundness:** 3
**Presentation:** 3
**Contribution:** 2
**Rating:** 4
**Confidence:** 4

**Summary:**

Authors propose a framework to use LLMS for synthesis planning. The key contribution from the paper is the mapping from a synthetic plan into a structured AND-OR representation. In this framework, LLMS are used only at the generation stage for applying single step transformations, this is then mapped into the AND-OR structured and expansions are selected using UCB criteria.
They use several techniques like RAG to improve performance and tun thorough evaluations on multiple public benchmarks.

**Strengths:**

Good and thorough evaluation on multiple benchmarks showing that this method can overperform other baselines in the literature. A lot of ablations that show the individual contibution of each component.

**Weaknesses:**

As I understand, the method relies on the LLMs to generate the molecule smiles and thus the chemical tranformations. This makes it a template free method and brings the question of what is the chemical validity of the proposed reactions.
The authors should discuss some specific examples and explore if the proposed reactions are chemically sensible.
There's also severals ways of doing this in large benchmark-scale as well:
Use atom-mapping tools or reaction analysis tools to discover if the proposed reactions are valid or not, or analize what reaction types are proposed?

- The use of RAG always bring the question of data leakage. This is a sensible topic that was not discussed on the paper. Given how much a performance gain RAG brings, even with only one sample, it makes sense to further study the reasons behind such a performance boost. Two possible ideas are:
1. Randomize the reactions given to the LLM, and measure performance. By doing this, authors could prove that the data being given as part of the RAG is indeed useful, and the boost is not due to some artifact of prompting.
2. A more controlled variation of this would be to fix the retrieved samples to the same set. This converts RAG into in-context learning and it works also as a good ablation.

-Analyze the overlap between the datasets being used for retrival, and the molecules and reactions in the benchmark.
- Use out of distribution examples and analyze then manually, showing actual molecular depictions so that it becomes more transparent for readers if the chemistry actually makes sense, beyond simple solve rates.

**Questions:**

See the weaknesses.

---

> ### Author Response · Authors · 2025-11-16
> **Response to Reviewer tQzg:**
>
> **Response to Reviewer tQzg:**
>
> We thank the reviewer for the thorough evaluation and constructive suggestions. We address each concern below.
>
> **Chemical Validity:** We would like to clarify an important misunderstanding: our method is not template-free. As detailed in Section 3.3.1 and Appendix B.1.4, every LLM-generated reaction undergoes rigorous template-based validation against the USPTO reaction database containing over 270,000 verified reaction templates. Only reactions matching known chemical transformations are accepted into the AND-OR tree. The LLM's role is to propose strategically coherent multi-step pathways, not to invent novel chemistry. This validation mechanism ensures systematic chemical validity guarantees throughout the search process.
>
> **Data Leakage Concern & Out-of-Distribution Analysis:**  As detailed in Section 4.1, we use established benchmark datasets following **Wang et al. (2025)**: USPTO-Easy, USPTO-190, Pistachio Reachable, and Pistachio Hard, with proper train/test separation.  For the retrieval database, we also explicitly construct it "from training and validation sets of USPTO with no overlap with test molecules" follows established practices in **Wang et al. (2025)** (as stated in our experimental setup, Section 4.1). We verified zero target molecule overlap between our test sets and the retrieval database.
>
> For distribution concern, we further provided retrieval statistics for our two most challenging test benchmarks. For USPTO-190, Tanimoto similarity distribution shows mean=0.65, median=0.63, with 44% exceeding 0.7 similarity, consistent with sampling from the same chemical space as the training set. For Pistachio Hard, the distribution shows mean=0.52, median=0.49, with 74% of molecules having max similarity below 0.6 to retrieved examples, indicating that the natural product targets are structurally distinct from typical USPTO molecules and represent more challenging synthesis scenarios. Our results demonstrate RAG's effectiveness across these difficulty levels, with strong performance maintained on Pistachio Hard despite the lower structural similarity to the retrieval database.
>
> **Randomized RAG Ablation:** We conducted the suggested experiment comparing similarity-based retrieval against random retrieval, both using 3 examples, with DeepSeek-V3 in 100 iterations. The table below shows that random retrieval degrades performance by 4-8% across iterations, confirming that strategic similarity, rather than prompt augmentation artifacts, drives RAG effectiveness:
>
> | Iterations | Pistachio Hard |  | USPTO-190 |  |
> | --- | --- | --- | --- | --- |
> |  | RAG | Random RAG | RAG | Random RAG |
> | 20 | 58.0 | 52.0 | 56.3 | 51.6 |
> | 40 | 71.0 | 65.0 | 72.1 | 67.9 |
> | 60 | 78.0 | 71.0 | 81.6 | 76.3 |
> | 80 | 83.0 | 75.0 | 85.3 | 79.5 |
> | 100 | 86.0 | 78.0 | 86.3 | 81.1 |
>
> We appreciate the reviewer's thorough engagement and believe these clarifications address the raised concerns. We will include the additional experimental results discussed above in the revised manuscript.
>
> Reference:
>
> - Wang, Haorui, et al.(2025) "LLM-Augmented Chemical Synthesis and Design Decision Programs." *International Conference on Machine Learning*.

---

### Author Response · Authors · 2025-11-16
**General Response (1/3)**

We sincerely thank all reviewers for their time and insightful feedback. We are encouraged that reviewers found our experimental evaluation to be "good and thorough" (tQzg, f8dE), our problem formulation "innovative" (6pLX), and our efficiency gains "substantial and practical" (f8dE). We have carefully considered all comments and believe they will help us significantly improve the clarity and impact of our paper.

We will first address some central themes and common questions raised by the reviewers, including clarifying our core contributions, addressing a key misunderstanding about chemical validation, and discussing the unique challenges of multi-step synthesis planning.

**1. Core Contribution: Bridging LLM Reasoning with Systematic Search**

Several reviewers (notably ZZ8L and f8dE) asked for a clearer articulation of our core technical novelty. We apologize if this was not sufficiently clear and would like to re-emphasize it here.

**The long-standing challenge** in multi-step retrosynthesis is the combinatorial explosion of the search space. Traditional search methods build the synthesis plan one step at a time, which is methodical but lacks strategic foresight. Recent LLM-based planners (e.g., LLM-Syn-Planner) leverage the remarkable chemical reasoning of LLMs to generate *entire* synthesis pathways at once. However, they typically use unstructured search methods like evolutionary algorithms. This approach is inefficient because it treats each candidate pathway in isolation, failing to learn from common sub-problems. It repeatedly re-discovers routes to common chemical intermediates, wasting expensive LLM inference calls.

**Our key innovation and contribution** is a framework that resolves this tension. AOT* is the first to synergistically integrate **holistic, multi-step pathway generation** from LLMs with the **systematic, memory-based exploration** of a classical AND-OR tree search. The novelty lies in the **atomic mapping** of entire, coherent synthesis plans generated by the LLM onto the tree structure.

This is fundamentally different from prior work:

- Unlike traditional methods (e.g., Retro*) that use a neural network as a simple one-step predictor, we use the LLM for its strategic, multi-step planning capability.
- Unlike other LLM planners (e.g., LLM-Syn-Planner), we don't discard the rich structural information of the search space. The AND-OR tree acts as a "memory", allowing AOT* to **cache and reuse successful routes** (i.e., solved intermediates) across all explored branches.

This hybrid approach directly addresses the core challenge: it leverages the LLM's chemical intelligence to make large, strategic leaps in the search space, while the tree structure ensures the search is systematic, avoids redundant work, and efficiently builds upon past discoveries. This synergy is the engine behind the 3-5x efficiency gains we demonstrate.

**2. A Critical Clarification: Ensuring Chemical Validity**

Reviewer **tQzg** raised a crucial point about the chemical validity of the reactions, operating under the assumption that our method was "template free." We apologize for the confusion and would like to clarify this critical point: **Our framework is not template-free.**

As detailed in Section 3.3.1 and Appendix B.1.4, every single reaction in an LLM-generated pathway undergoes a rigorous **template-based validation process**. We match each proposed transformation against a database of over 270,000 known reaction templates derived from the USPTO dataset. A reaction is only accepted and added to the AND-OR tree if it corresponds to a known, valid *chemical transformation*. This grounding step is essential to our method; it constrains the LLM's creativity to the space of established chemistry, preventing hallucinated or chemically invalid steps and ensuring the final synthesis plans are sound.

---

> ### Author Response · Authors · 2025-11-16
> **General Response (2/3)**
>
> **3. On the Power and Grounding of LLMs in Chemistry**
>
> Reviewer **6pLX** asked for the intuition behind why a general LLM, without specific fine-tuning, can perform complex chemical reasoning. This is an excellent question that touches upon the emergent capabilities of modern foundation models. The capacity for sophisticated chemical reasoning is not explicitly programmed but rather emerges from the model's training on colossal, internet-scale datasets. This data contains a vast repository of human chemical knowledge, including:
>
> - **Scientific publications and textbooks:** Decades of chemical literature from journals (e.g., ACS, RSC), textbooks, and patents are embedded in the training corpus.
> - **Public chemical databases:** Information from websites like PubChem, Wikipedia, ChEMBL is also ingested.
>
> Molecules are often represented textually using SMILES strings, and reactions can be described as **`reactants>>products`**. The LLM learns the deep grammatical and semantic patterns of this "language of chemistry" just as it learns a natural language. This allows it to understand concepts like functional groups, reaction mechanisms, and retrosynthetic logic from context.
>
> This emergent capability has been systematically documented in recent literature. For instance, **Guo et al. (2023)** conducted a comprehensive benchmark on eight chemical tasks and found that general-purpose LLMs like GPT-4 "possess significant chemical knowledge and reasoning ability" even in zero-shot settings, without any domain-specific fine-tuning. Similarly, **White et al. (2023)** assessed the chemical knowledge of large language models, concluding that their strong performance stems from being "exposed to a huge corpus of text including textbooks, scientific articles, and chemical websites." Their work demonstrates that these models can often match or even exceed the performance of purpose-built, fine-tuned models on certain tasks. A recent review by **Jablonka et al. (2024)** further surveys this phenomenon, noting that LLMs can serve as a powerful "knowledge base for predictive chemistry" due to the extensive scientific information they absorb during pre-training.
>
> Furthermore, our use of **Retrieval-Augmented Generation (RAG)** is vital for steering this general knowledge toward a specific problem. As Reviewer **tQzg** correctly noted, RAG provides a significant performance boost. This is because it grounds the LLM at inference time, providing it with specific, validated synthesis routes for molecules structurally similar to the current target. This acts as a powerful form of in-context learning, guiding the LLM to apply its vast, generalized knowledge to propose more plausible and effective strategies for the specific task at hand. We have taken care to prevent data leakage by ensuring the retrieval database has no overlap with our test sets (Sec 4.1).
>
> ### **4. On Efficiency, Cost, and Fair Comparisons**
>
> Reviewer **f8dE** made an excellent point regarding the definition of an "iteration" and the importance of comparing wall-clock time and monetary cost.
>
> - **Why "Iterations" is a Fair Metric:** We argue that an "iteration" is a fair and primary metric for *search efficiency*. In both AOT* and our main LLM-based baseline (LLM-Syn-Planner), an iteration corresponds to a single call to the core generative module—the LLM—to produce one or more complete pathways. It, therefore, measures how many generative steps are needed to find a solution, directly reflecting the efficiency of the overarching search strategy. Our results consistently show that AOT*'s systematic exploration requires far fewer of these generative steps.
> - **Cost and Time Analysis:** We completely agree that a cost-based analysis is crucial for a practical perspective. We have included a detailed cost-performance analysis in **Table 5 and an extended version in Table 13 of the Appendix**. These tables show that AOT* is not only more iteration-efficient but also more cost-effective. For instance, AOT* paired with the moderately-priced DeepSeek-V3 outperforms the baseline LLM-Syn-Planner paired with the much more expensive GPT-4o. This demonstrates that our algorithmic improvements provide tangible, real-world benefits, enabling high performance even with more economical models.
>
> We appreciate the reviewer's feedback and would like to clarify our method's technical novelty by pointing to the key sections where our approach is systematically detailed. The specific problem formulation is located in Section 3.1. The technical implementation of our novel pathway-to-tree mapping is described in Section 3.2 and Algorithm 1, with its theoretical underpinnings discussed in Section 3.4. Our extensive experiments further validate the robustness of this framework. We acknowledge that the connections between these sections could be clearer, and we will revise the text to ensure our core contributions are more readily apparent to the reader.

---

> > ### Author Response · Authors · 2025-11-16
> > **General Response (3/3)**
> >
> > We believe these clarifications address the most critical questions raised by the reviewers.
> > We are confident that our work represents a significant step forward in making LLM-powered synthesis planning more efficient and practical. We thank the reviewers again for their constructive engagement, which will undoubtedly help us strengthen the final version of our manuscript.
> >
> > **References:**
> >
> > - Guo, T., et al. (2023). What can large language models do in chemistry? a comprehensive benchmark on eight tasks. *Advances in Neural Information Processing Systems*.
> > - White, A. D., et al. (2023). Assessment of chemistry knowledge in large language models that generate code. *Digital Discovery*.
> > - Jablonka, K. M., et al. (2024). Leveraging large language models for predictive chemistry. *Nature Machine Intelligence*.

---

### Note · Authors · 2026-01-04

I have read and agree with the venue's withdrawal policy on behalf of myself and my co-authors.